# The naive T-cell receptor repertoire has an extremely broad distribution of clone sizes

Peter C de Greef[1†], Theres Oakes[2†], Bram Gerritsen[1,3†], Mazlina Ismail[2], James M Heather[2], Rutger Hermsen[1], Benjamin Chain[2*], Rob J de Boer[1*]

[1]Theoretical Biology and Bioinformatics, Utrecht University, Utrecht, Netherlands; [2]Division of Infection and Immunity, University College London, London, United Kingdom; [3]Department of Pathology, Yale School of Medicine, New Haven, United States

**Abstract** The clone size distribution of the human naive T-cell receptor (TCR) repertoire is an important determinant of adaptive immunity. We estimated the abundance of TCR sequences in samples of naive T cells from blood using an accurate quantitative sequencing protocol. We observe most TCR sequences only once, consistent with the enormous diversity of the repertoire. However, a substantial number of sequences were observed multiple times. We detect abundant TCR sequences even after exclusion of methodological confounders such as sort contamination, and multiple mRNA sampling from the same cell. By combining experimental data with predictions from models we describe two mechanisms contributing to TCR sequence abundance. TCRα abundant sequences can be primarily attributed to many identical recombination events in different cells, while abundant TCRβ sequences are primarily derived from large clones, which make up a small percentage of the naive repertoire, and could be established early in the development of the T-cell repertoire.

*For correspondence:
b.chain@ucl.ac.uk (BC);
R.J.DeBoer@uu.nl (RJB)

†These authors contributed equally to this work

**Competing interests:** The authors declare that no competing interests exist.

## Introduction

The human adaptive immune system employs a vast number (> $10^{11}$ [*Clark et al., 1999*]) of T lymphocytes, to detect and control pathogens. Most T cells express a single T-cell receptor (TCR) variant, which binds antigen in the form of a short peptide presented by the Major Histocompatibility Complex (pMHC) (*Davis and Bjorkman, 1988*). The TCR has to be specific to distinguish between self- and non-self-pMHC, but due to the large number of possible foreign antigens (> $20^9$) a specific TCR is nevertheless expected to bind many different pMHC (i.e., cross-reactivity) (*Mason, 1998*; *Sewell, 2012*). The actual diversity of the TCR repertoire is unknown, but with improved sequencing techniques, estimates have risen by orders of magnitude from $10^6$ (*Arstila et al., 1999*), $10^7$ (*Robins et al., 2009*), to over $10^8$ (*Qi et al., 2014*).

Generation of αβ TCRs occurs in the thymus, where thymocytes randomly rearrange and imprecisely recombine gene segments to create a complete receptor (*Nikolich-Zugich et al., 2004*). This heterodimer is generated by random recombination of Variable, Diversity, and Joining (V, D and J) segments for TCRβ, and V and J segments for TCRα sequences (*Davis and Bjorkman, 1988*). Most variability arises due to random nucleotide insertions and deletions where the segments are joined (*Murugan et al., 2012*). Recent estimates of the potential number of TCRs produced by this V(D)J-recombination process range from > $10^{20}$ (*Zarnitsyna et al., 2013*) to $10^{61}$ (*Mora and Walczak, 2019*), which vastly outnumbers the number of distinct TCRs present in a human body. After generation of the TCR, T cells undergo positive and negative selection, which selects those T cells that have sufficient, but not too high, affinity for any self-pMHC (*McDonald et al., 2015*). About 3–5% of

thymocytes survive selection (*Merkenschlager et al., 1997*) and enter the periphery as T cells that have not yet encountered foreign cognate antigen, that is as naive T cells.

The thymic output of new T cells decreases because of thymic involution, making peripheral division of existing cells the main source of naive T cells from early adulthood onwards in humans (*den Braber et al., 2012*; *Kumar et al., 2018*). In the periphery, naive T cells compete for cytokines, such as IL-7, and need to interact with self-pMHC to survive (*Tanchot et al., 1997*; *Takada and Jameson, 2009*; *Jenkins et al., 2010*). Competition between T-cell specificities may reduce repertoire diversity when cells with some TCRs outcompete others (*De Boer and Perelson, 1994*), resulting in differences in TCR frequencies, and heterogeneous naive T-cell clone sizes. Experimental evidence for large heterogeneity in division and survival rates within the naive T-cell pool has been shown in mice (*Hogan et al., 2015*; *Rane et al., 2018*; *Reynaldi et al., 2019*). Such experiments are not feasible in humans, but mathematical modelling has been used to assess how fitness differences between T-cell clones may affect the frequency of clones in the naive repertoire (*Stirk et al., 2008*; *Stirk et al., 2010*; *Hapuarachchi et al., 2013*; *Lythe et al., 2016*; *Desponds et al., 2016*; *Desponds et al., 2017*; *Dowling and Hodgkin, 2009*; *Johnson et al., 2012*).

Measuring the distribution of TCRα and TCRβ sequences in samples of naive T cells can inform us about the clone-size distribution of the naive T-cell repertoire. Previous studies have reported large heterogeneity in the frequency of TCRβ sequences in naive repertoires from mice (*Quigley et al., 2010*) and humans (*Robins et al., 2009*; *Venturi et al., 2011*; *Qi et al., 2014*; *Pogorelyy et al., 2017*). One important factor shaping the abundance of TCR sequences is their likelihood to be produced during VDJ-recombination. Rearrangements with less N-insertions, for example, tend to be more commonly observed (*Robins et al., 2009*; *Robins et al., 2010*; *Venturi et al., 2011*; *Pogorelyy et al., 2017*). To study this in more detail, the Mora and Walczak groups developed probabilistic models that predict the generation probability of any specific TCRα or TCRβ sequence (*Murugan et al., 2012*; *Marcou et al., 2018*). They showed that these sequences ($\sigma$) differ by several orders of magnitude in their probability $\mathcal{P}(\sigma)$ of being produced by V(D)J recombination in the thymus. Differential generation probabilities do not only impact the abundance of TCRα and TCRβ sequences within an individual, but also contribute to sharing among individuals (*Robins et al., 2010*; *Quigley et al., 2010*; *Venturi et al., 2011*; *Qi et al., 2014*; *Pogorelyy et al., 2017*; *Elhanati et al., 2018*). Hence, it is essential to take the likelihood of generating a sequence into account when interpreting sequencing data of immune repertoires.

In this study, we characterize the frequency distribution of TCRα and TCRβ sequences in the naive repertoire. We analyze published and new experimental data on both the TCR α and β chain, and combine a quantitative unique molecular identifier (UMI)-based TCR sequencing pipeline with mathematical modeling to consider carefully the contributions of different mechanisms that may lead to observed abundant TCRα and TCRβ sequences in the naive repertoire. Such mechanisms include experimental confounders, such as the purity of the cell populations and repeated sampling of mRNA from the same cell, and diverse biological processes including distinguishing carefully between repeat generation of identical sequences in different cells, and large naive T-cell clones. We show that all these processes are likely to contribute to the observed abundance profile of TCR sequences in samples of naive repertoires. In particular, even after all other mechanisms are accounted for, we find evidence for naive T-cell clone size heterogeneity. Specifically, the results are compatible with an underlying power-law distribution of naive T-cell clone sizes (*Desponds et al., 2016*), or more generally by models in which 1–5% of naive T cells represent large clones of $10^5$ - $10^6$ cells. Preferential expansion of some clonotypes, perhaps those occurring early in development of adaptive immunity, therefore plays an important role in shaping the naive T-cell repertoire.

## Results

We analysed the frequency distribution of TCR sequences in the naive T-cell compartment, using TCRα and TCRβ sequences published in *Oakes et al. (2017)*. In brief, peripheral blood mononuclear cells (PBMCs) from two adult volunteers were FACS-sorted into naive (CD27$^+$CD45RA$^{high}$) and various memory CD4$^+$ and CD8$^+$ populations. TCRα and TCRβ mRNA was reverse transcribed to cDNA molecules to which unique molecular identifiers (UMIs) were attached, followed by PCR-amplification and high-throughput sequencing (HTS) on an Illumina MiSeq platform. We refer to this as experiment 1 below (for further details see Section 'Sequence analysis'). Sequence reads were processed

using a customized version of the Decombinator pipeline (*Thomas et al., 2013*), with an improved error correction on UMIs to more reliably estimate the frequency of nucleotide TCRα and TCRβ sequences in the samples (see Section 'Sequence analysis'). Additionally, we used the RTCR pipeline (*Gerritsen et al., 2016*) for comparison (Section 'Sequence analysis'). The different memory populations were combined for the purpose of the analysis presented below.

## Abundant TCR sequences are frequently shared between naive and memory populations, and are enriched for high V(D)J recombination probabilities

Within the naive T-cell repertoires, the vast majority of TCRα and TCRβ sequences were observed only once, and most frequencies fall within the range from 1 to 5 (*Figure 1A*). As expected, in the memory repertoires, which contain clonally expanded T cells, much more abundant sequences were present, with a substantial number of α and β chains observed more than 1000 times (*Figure 1A*). The few sequences observed with a frequency higher than five in the naive samples were shared in most cases (94.6%) with the corresponding memory subset from the same individual. We examined whether this overlap might arise from imperfect sorting of the T-cell populations, despite the tight non-overlapping sort gates applied (see [*Oakes et al., 2017*]). A prediction of such sorting contamination is that the abundance of the shared TCR sequences in the naive and memory repertoires should be proportional. Such a linear relationship could be observed clearly for CD8$^+$ TCRα and TCRβ sequences (*Figure 1A*), especially for memory abundances greater than 1000. Correlation measurements suggested that the amount of contamination for CD8$^+$ T cells was 0.1 - 1.5%. As expected, no correlation was observed between the abundance of TCR sequences shared between naive and memory populations of different donors (*Figure 1B*).

We next examined the relationship between V(D)J recombination probabilities and the overlap between naive and memory repertoires. Using the V(D)J-recombination model of *Marcou et al. (2018)*, we predicted the generation probabilities $\mathcal{P}(\sigma)$ of all TCRα and TCRβ sequences in our datasets. As expected, we observed a wide range of $\mathcal{P}(\sigma)$ values, which were several orders of magnitude higher for TCRα sequences than TCRβ, due to additional recombination of the D segment. The generation probability distributions of sequences derived from naive and memory T cells were indistinguishable (*Figure 1C*, blue and red, respectively). Thus, our data provide no evidence that the V(D)J-recombination process preferentially produces sequences that are more likely to enter the memory pool during an immune response. However, TCRα sequences shared between memory and the corresponding naive samples, were strikingly enriched for high $\mathcal{P}(\sigma)$ (*Figure 1C*, green). This enrichment is much less evident for TCRβ sequences. The enrichment for sequences with high $\mathcal{P}(\sigma)$ in the population of shared memory/naive TCRα is not compatible with overlap derived from contamination during cell sorting, but rather suggests that the sharing may also arise from T cells which use the same TCRα because of identical VJ recombination events in different T cells. It is important to stress that, since such different T cells are highly unlikely to also share TCRβ sequences, the clonotype, and hence specificity of the T cells in the naive and memory compartments may well be different, despite sharing TCRα sequences.

As a control, we also analyzed overlap between the naive sample from one volunteer and the memory sample from the other. In this case, sort contamination of naive repertoires by memory T cells is excluded and a shared sequence can only result from independent identical recombination events, from distinct T-cell clones. For CD4$^+$ cells, we find that the number of TCRα sequences shared between naive and memory is similar between and within volunteers, and that the $\mathcal{P}(\sigma)$ distribution is nearly identical (*Figure 1C*, purple). For CD8$^+$ cells, the number of sequences shared within an individual is somewhat larger than between individuals, compatible with some degree of sort contamination in this population as discussed above. The small number of TCRβ sequences shared between individuals also had a relatively high $\mathcal{P}(\sigma)$, although considerably smaller than for TCRα.

In summary, although contamination with abundant memory T cells may make a small contribution to the TCR sequences which are found in both naive and memory for CD8$^+$ cells, multiple identical recombinations arising from high $\mathcal{P}(\sigma)$ values is the dominant mechanism leading to overlap in the TCRα repertoires. Nevertheless, in order to stringently exclude any possible contribution of contamination, we included an analysis which excluded all the shared sequences from the further investigations of the relationship between TCR sequence abundance and $\mathcal{P}(\sigma)$ (*Figure 1D*).

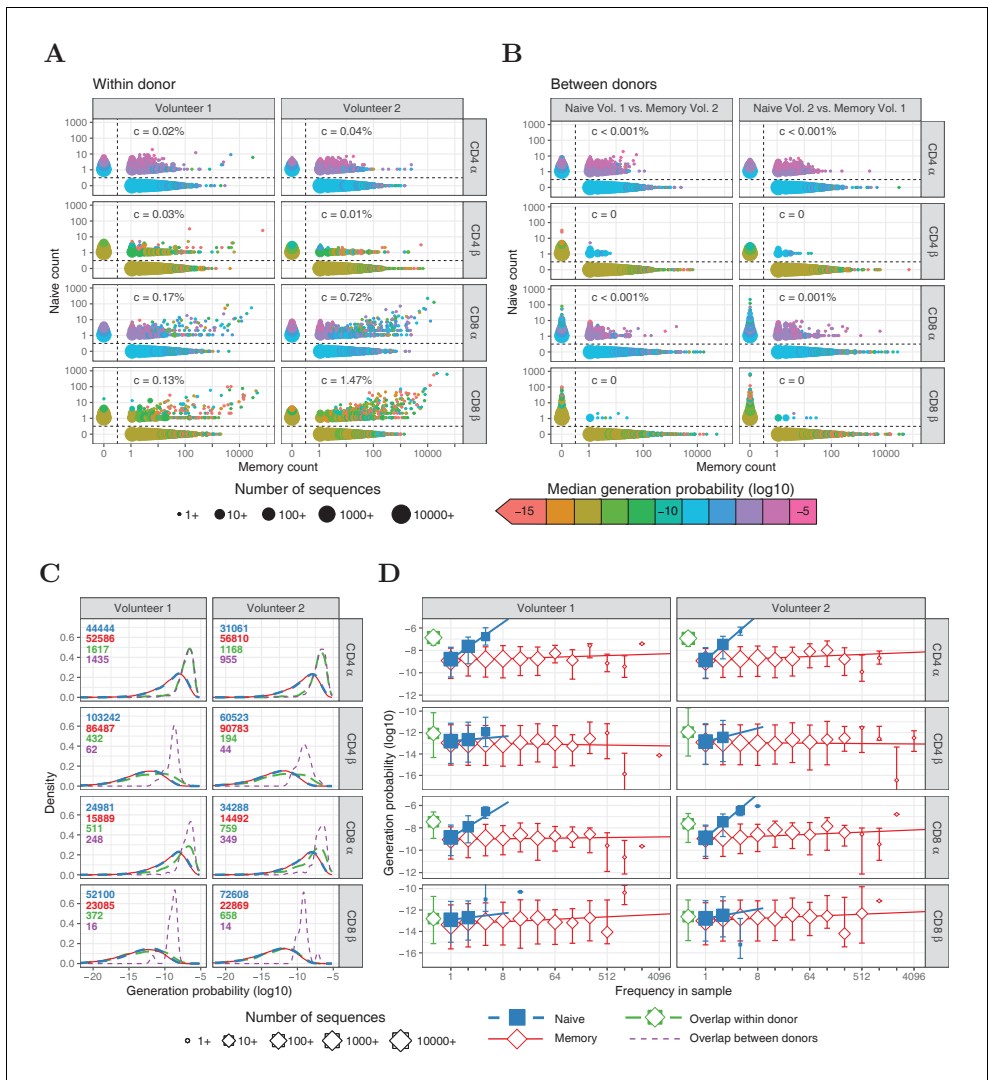

**Figure 1.** Frequencies and generation probabilities of TCRα and TCRβ sequences from memory and naive T cells. (A) Frequency of TCRα and TCRβ sequences in naive versus total frequency in memory repertoires sampled from the same volunteer. Symbol sizes represent number of sequences with these frequencies and colour represents their median generation probability $\mathcal{P}(\sigma)$, as determined using IGoR (**Marcou et al., 2018**). The $c$ value is the slope of linear regression on sequences with a memory count > 100 and indicates the estimated probability that a given TCR sequence from a memory cell appears in the naive sample. (B) As A., but comparing frequency in naive sample from one volunteer with frequency in memory from the other volunteer. (C) Distributions of generation probabilities (log10) for TCR α and β sequences from CD4[+] and CD8[+] from two volunteers. Blue dashed: naive, red solid: memory, green long-dashed: overlap (i.e., sequences observed in both naive and memory within a volunteer), purple dashed: overlap between volunteers (i.e., sequences observed in the naive subset of Volunteer 1 and a memory subset of Volunteer 2, or vice versa). The total number of sequences for each group are indicated in corresponding colors. (D) The median $\mathcal{P}(\sigma)$ is shown for each observed frequency class (log2 bins) of sequences exclusively observed in naive (blue squares) or memory T-cell (red diamonds) samples. $\mathcal{P}(\sigma)$ of the overlapping chains is shown in green for reference (irrespective of frequency). Symbol sizes indicate numbers of sequences for each frequency class. Error bars represent the 25% and 75% quartiles, solid lines indicate linear regression between observed frequency and $\mathcal{P}(\sigma)$, weighted by the number of sequences with that frequency.

The online version of this article includes the following source data and figure supplement(s) for figure 1:

**Source data 1.** Memory and naive counts in Experiment 1.
**Figure supplement 1.** TCRα and TCRβ sequences abundant in naive tend to have less N-insertions.
**Figure supplement 2.** Similar to *Figure 1*, but for HTS data processed with RTCR.

The abundances of sequences in all naive repertoires were correlated to $\mathcal{P}(\sigma)$ (*Figure 1D*, blue). The median $\mathcal{P}(\sigma)$ of the α chains that were observed at least three times was about 154-fold higher than for those that have only been observed once ($p<10^{-15}$, Wilcoxon test). The enrichment for high $\mathcal{P}(\sigma)$ in more abundant TCR sequences was weaker for TCRβ (~2.5-fold, $p<0.01$, Wilcoxon test), but still stronger than for memory subsets (1.65- and 1.03-fold for TCRα and TCRβ, respectively, $p<10^{-15}$ and $p = 0.27$). In line with this, the number of N-additions tended to be lower for TCRα and TCRβ sequences abundant in the naive samples (*Figure 1—figure supplement 1*). These correlations suggest that multiple identical recombination events which occur during formation of the naive T-cell repertoire in the thymus due to high generation probabilities, contribute to the observation of abundant TCR sequences. This is especially evident for TCRα, where the probabilities of producing a given sequence are higher because of the absence of a D region. However, abundant TCR sequences with low $\mathcal{P}(\sigma)$ are also observed, especially for TCRβ, leaving open the possibility of large naive T-cell clones.

## Frequently observed TCR sequences cannot be attributed only to multiple RNA molecules per cell

T cells contain on average in the order of 100 molecules of TCRα and 300 molecules of TCRβ mRNA (*Oakes et al., 2017*). Because the TCR sequencing pipeline is not 100% efficient, only a small proportion of these molecules are actually sequenced, but the possibility remains that TCR sequences observed multiple times may be due to repeat sampling from the same cell. Because the variance of this number remains undetermined, it is difficult to computationally determine the contribution of this multiple sampling to the data. Instead, we performed an additional experiment (referred to as experiment 2) in which we sorted naive T cells from an additional volunteer, and split the naive T cells into three subsamples before mRNA extraction. We then carried out library preparations and sequenced TCRα and TCRβ sequences from each subsample independently. In this experiment, sequences observed in more than one subsample must have been derived from different cells, and cannot be a result of sequencing multiple mRNA molecules from a single cell. Repeated sequences must therefore derive from different cells, and represent abundant sequences.

In total 16913 (3.4%) TCRα sequences, and 5744 (0.61%) TCRβ sequences, were observed in more than one subsample (*Figure 2A*), confirming the existence of a substantial number of frequent TCR α and β chains in the full naive repertoire. In order to exclude any contribution from sort contamination, we also plot the data after removing all TCR sequences found in both memory and naive repertoires (*Figure 2A*, grey bars). A substantial number of α and β chains were still found in multiple subsamples. In order to estimate the impact of multiple sampling on the observed abundances we randomly permuted the TCR sequences between subsamples, and reanalyzed the distributions (see detailed explanation in Section 'Subsampling to exclude inflated abundance through multiple RNA contributions by single cells'). We estimated that ~25% of α and > 75% of β chains with an abundance of greater than 1 in an individual sample may arise from sampling multiple RNA molecules from single cells. The impact is strongest on TCR sequences observed twice (see *Figure 2—figure supplement 1*). Thus multiple mRNA sampling is an important confounder of estimating TCR sequence abundances in individual repertoires, especially for TCRβ.

Having ruled out the contribution of multiple mRNA sampling experimentally, we examined the relationship between TCR sequence abundance and $\mathcal{P}(\sigma)$ in this new data set. The TCRα chains present in more than one naive subsample are dominated by sequences with high $\mathcal{P}(\sigma)$. The median generation probability of TCRα sequences observed in two and three subsamples was 56- and 165-fold higher, respectively, than those observed only once (*Figure 2B*). The relationship for TCRβ sequences was remarkably different, however. While TCRβ sequences observed in two subsamples are mildly enriched for high generation probabilities, those observed in three subsamples have hardly any enrichment for high $\mathcal{P}(\sigma)$ (*Figure 2B*). Instead, their generation probabilities tend to be lower than those of the sequences observed in two subsamples, and more similar to the generation probabilities of TCRβ sequences seen in only one subsample. We obtained similar results when measuring the number of N-additions and VJ-deletions in the rearrangements: abundant α chains (with incidence 2 or 3) tend be closer to germline rearrangements, while this was only the case for β chains with incidence 2, and not for the most abundant β chains with incidence 3 (*Figure 2C and D*). These trends were observed both with and without removing the sequences that were also observed in

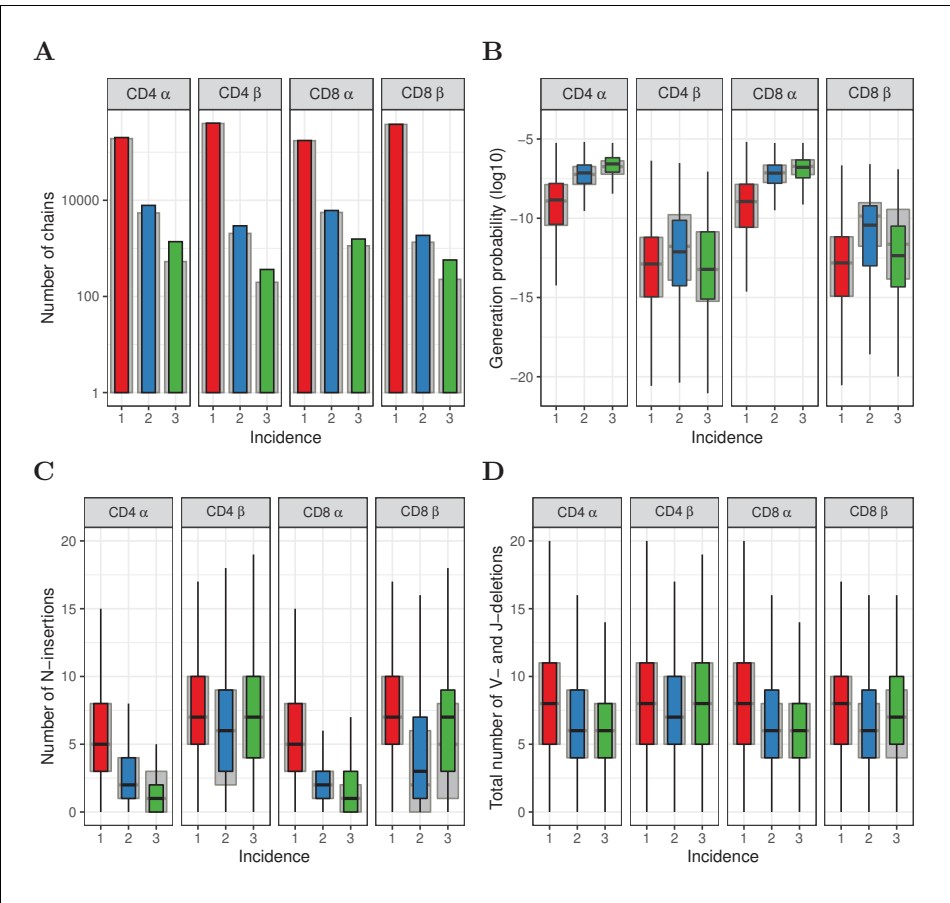

**Figure 2.** Subsampling naive T cells confirms that frequently observed TCRα but not TCRβ sequences have high generation probabilities. (**A**) The number of TCRα and TCRβ sequences observed in 1, 2 or 3 subsamples (experiment 2). The grey background bars show the results after removing all sequences that were also observed in the corresponding memory samples. (**B**) Generation probabilities $\mathcal{P}(\sigma)$ (log10) of TCRα and TCRβ sequences observed in 1, 2 or 3 subsamples. (**C**) Minimal number of N-additions of TCRα and TCRβ sequences observed in 1, 2 or 3 subsamples. (**D**) Number of V- and J-deletions of TCRα and TCRβ sequences observed in 1, 2 or 3 subsamples. The plot shows median (black horizontal line), interquartile range (filled bar) and the range from the bar up to 1.5 times the interquartile range (black vertical range, outliers not shown).

The online version of this article includes the following source data and figure supplement(s) for figure 2:

**Source data 1.** Naive TCRα and TCRβ abundance in the subsamples of Experiment 2.
**Figure supplement 1.** Permutation of subsampling experiment.
**Figure supplement 2.** Similar to *Figure 2*, but for HTS data processed with RTCR.
**Figure supplement 3.** Observed frequency predicts sharing for TCRα but not TCRβ sequences.

memory (*Figure 2*, grey versus colored bars) and when processing the data with RTCR (*Figure 2— figure supplement 2*).

We further explored whether the more abundant sequences were also more 'public' (found in the repertoires of multiple individuals), which would be predicted if they are more likely products of V(D) J-recombination. We measured the degree of sharing between those TCR sequences observed in 1, 2, or 3 naive subsamples, and the TCRα and TCRβ repertoires of unfractionated blood samples collected from 28 healthy donors (details in Section 'Sharing of TCRα and TCRβ sequences'). Both TCRα and TCRβ sequences observed in two or three subsamples were found to be significantly more often shared with this independent cohort than those observed once (*Figure 2—figure supplement 3A*). The most frequent TCRα sequences, which were seen in three subsamples, showed the highest sharing degree, consistent with their strongest enrichment for high generation probabilities. The relatively small number of most frequent TCRβ sequences (i.e., those observed in three

subsamples), did not show increased inter-individual sharing compared to the TCRβ sequences observed in two subsamples. Additional comparison with publicly available TCRβ data from a large cohort (*Emerson et al., 2017*) (see Section 'Sharing of TCRα and TCRβ sequences') showed that the most frequently observed β chains, which were observed in all three subsets in experiment 2, were less public than sequences observed in two subsamples (*Figure 2—figure supplement 3B*). The seemingly paradoxical finding that the most abundant TCRβ sequences (observed in all three subsamples) have lower $\mathcal{P}(\sigma)$, and are less public than those found twice, is explored in more detail below.

## Computational models of TCR repertoire generation suggest the presence of a small proportion of large T-cell clones in the naive repertoire

In order to more rigorously test our ideas about the frequency distribution of clonotypes in the naive T-cell repertoire, we explored a number of possible computational models of repertoire generation and sampling, and compared model predictions with the experimental data discussed above. The first simplest scenario we considered was a neutral model of repertoire formation, similar to Hubbell's Neutral Community Model (*Hubbell, 2001*; *Figure 3A*, details in Section 'Neutral model for dynamics of naive T cells'). The model assumes that there is no selective advantage of one TCR over another, and therefore the TCR of a naive T cell does not affect its lifespan or division rate. Consider a pool of $N$ naive T cells, from which cells are removed by cell death or by priming with antigen, leading to differentiation into a memory population. A fraction θ of these cells is replaced by thymic production of new clones and the remaining fraction 1 - θ gets replaced by division of cells present in the pool. When simulating the naive T-cell pool with this model, the clone-size distribution approaches a 'steady state' (not shown). We use this steady-state distribution, for which we have an analytical expression (Section 'Neutral model for dynamics of naive T cells') to predict the size of clones in the naive T-cell pool. As the contribution of thymic output decreases during aging (*Steinmann et al., 1985*), we evaluated the model for a wide range of values for θ. The clone-size distribution which emerges from the neutral model is approximately geometric for clone sizes larger than the introduction size $c$ (*Figure 3B*, Section 'Neutral model for dynamics of naive T cells'). We compared this basic model to models in which we impose other distributions on the underlying clonotype abundances (model details in Section 'Clone-size distributions of the naive T-cell pools'). We specifically focused on heavy-tailed distributions such as log-normal and power-law distributions,

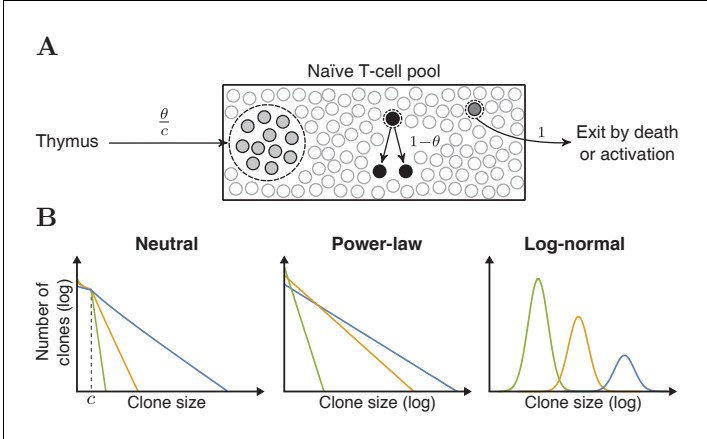

**Figure 3.** Schematic representation of the neutral model and various clone-size distributions. (**A**) Schematic representation of the dynamics of the neutral model for the naive T-cell pool. Each event starts with removal of one randomly selected cell from the pool, followed by peripheral division of another cell (with probability 1- θ), or a chance for thymic production (probability θ). After $c$ of these thymus events, a clone of $c$ cells is generated and added to the peripheral pool, reflecting the divisions of T cells before entering the periphery. (**B**) Schematic representations of the various clone-size distributions that were used to predict the naive repertoire. The green, orange and blue colored lines depict three parameter choices for each distribution, resulting in a low, medium and high mean clone size, respectively.

which have previously been associated with T-cell repertoires (*Desponds et al., 2016*). The shape of each of these distributions is controlled by a single parameter (as shown in *Figure 3B*), allowing us to compare distributions with different degrees of heterogeneity. In all cases, we normalized the clone-size distribution such that the total number of cells $N$ is constant. Since we had separate experimental data for CD4$^+$ and CD8$^+$ cells, we considered CD4$^+$ and CD8$^+$ cells separately, setting $N(CD4) = 7.5 \times 10^{10}$, and $N(CD8) = 2.5 \times 10^{10}$.

From all model clone-size distributions we simulate three subsamples, so as to compare with the data from the second experiment described above. Each sampled TCR is assigned a TCRα and a TCRβ sequence that were generated with IGoR (*Marcou et al., 2018*). Previous studies showed that α and β chains with higher generation probabilities tend to have a higher probability to survive selection (*Elhanati et al., 2014*). Therefore, we train a simple $\mathcal{P}(\sigma)$-dependent selection model on the data from the single naive T-cell samples shown in *Figure 1*. First, we assume that productively rearranged chains have an overall 1/3 probability to survive thymic selection. Then we bias the probability for bins of sequences based on their $\mathcal{P}(\sigma)$, such that the resulting set of α and β chains has the same generation probability distribution as in the experimental repertoire data (Section 'In silico samples from modelled clone-size distributions'). The models also incorporate the expected number of cells that contribute at least one mRNA molecule. This parameter is also learnt from the data, by setting the number of cells that contributed mRNA such that the predicted diversity of a subsample matches the observed diversity (Section 'In silico samples from modelled clone-size distributions'). Taken together, the subsamples we take from the various model clone-size distributions are such that they match the generation probabilities and diversity of the experimental subsamples as closely as possible. We compare the number and median $\mathcal{P}(\sigma)$ of the TCR sequences that are predicted to occur in only one, in two or in three subsamples, with the equivalent experimental data from experiment two above (*Figure 4*).

We consider first the neutral model (*Figure 4A and C*). For the α chain, a wide range of thymic output rates predict the number of chains occurring in 1, 2 and 3 subsamples reasonably well (*Figure 4A*). The model does not predict the median $\mathcal{P}(\sigma)$ of TCRα found in 2 and 3 subsamples well, although qualitatively the model does predict the increasing $\mathcal{P}(\sigma)$ with increasing abundance (*Figure 4C*). For the β chains, there is no range of thymic output rates for which the model correctly predicts the number of sequences observed in 2 and 3 subsamples. Moreover, the observation that incidence two chains have higher $\mathcal{P}(\sigma)$ than incidence three chains was not predicted for any value of θ (*Figure 4C*). Thus, although the neutral model captures some features of the observed TCRα sequence abundances, it cannot account for observed TCRβ distributions. A similarly poor match between observed and predicted data is observed for log-normal clonotype model (*Figure 3B*) distributions (not shown).

In contrast, there is a much better fit between observed and predicted data is obtained when the model clonotype frequencies are modelled by a power-law distribution (*Figure 4B and D*). Like the distributions discussed above, in the parameter range where clone-size heterogeneity is limited (i.e. a steep slope), a power-law distribution predicts both the number of TCRα sequences found in 2 and 3 samples, and their larger median $\mathcal{P}(\sigma)$. The number of TCRβ sequences is also predicted well if the slope is close to 2.3 (*Figure 4B*). Remarkably, for this slope the median $\mathcal{P}(\sigma)$ of TCRβ sequences found in two samples is higher than the median $\mathcal{P}(\sigma)$ of TCRβ sequences found in three samples (*Figure 4D*). Intuitively, we can understand this observation as reflecting the properties of power-law distributions, combined with the lower generation probabilities of TCRβ. Identical TCRβ recombinations occur frequently enough to make a detectable contribution to the TCR sequences observed in two samples, but not to those detected in three samples. Therefore, a significant proportion of TCRβ sequences observed twice are in fact derived from two or more different naive T-cell clones. In contrast, TCR sequences observed three times (or more) must be derived from large naive T-cell clones. Abundant TCRα sequences arise both from large clones and summation of identical TCRα from multiple smaller clones, but due to their higher generation probabilities, the latter dominates the $\mathcal{P}(\sigma)$ for TCRα sequences found twice and three times. Finally, we note that although TCR sequence abundance in the single samples from experiment one is likely to incorporate multiple mRNA from single cells, the power-law distribution also predicts abundances in the single samples of experiment one reasonably well (*Figure 4—figure supplement 1*).

The vast majority of TCR sequences in samples of naive T cells are observed only once, and hence we cannot infer anything about their frequency in the whole repertoire, except that it is likely to be

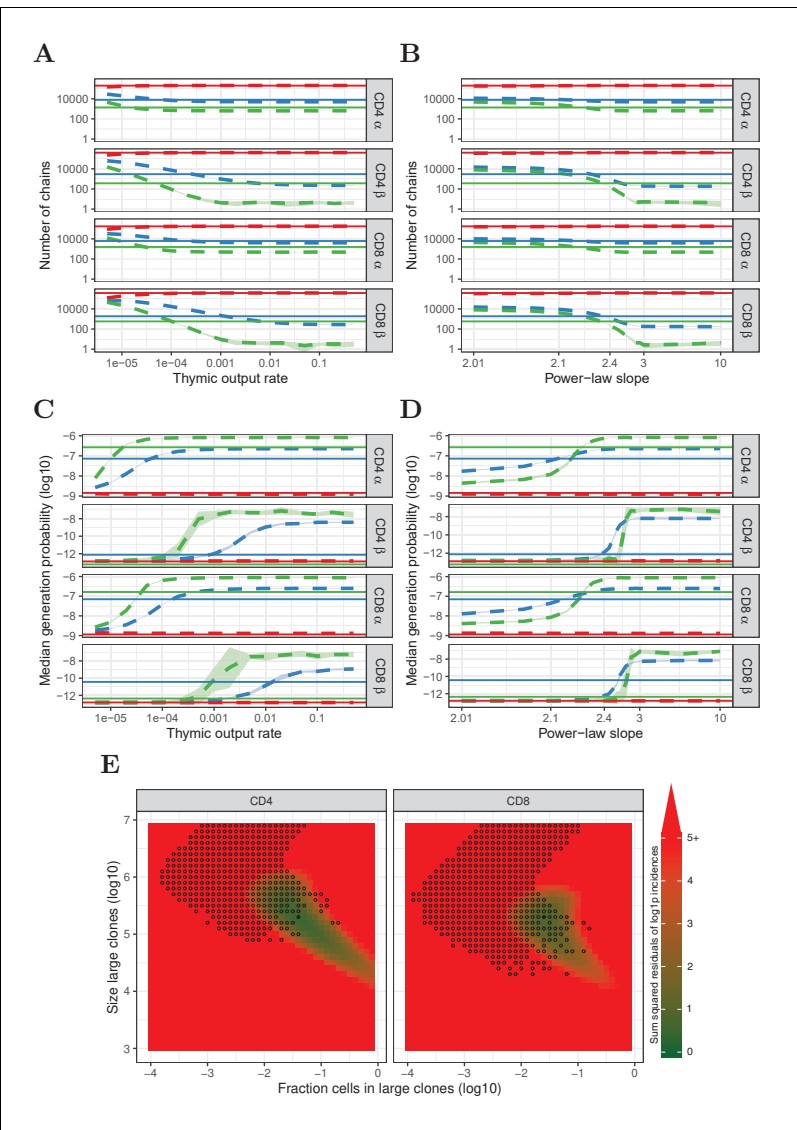

**Figure 4.** Predictions of the neutral, power-law and two-population model compared with HTS data. (**A**) Number of TCRα and TCRβ sequences which are predicted to be shared between 1 (red), 2 (blue) and 3 (green) subsamples as a function of the thymic output rate θ for the neutral model. (**B**) As A., but as a function of the slope of the power-law distribution. (**C**) The median generation probability $\mathcal{P}(\sigma)$ of TCRα and TCRβ sequences predicted by the neutral model. Dashed lines depict the mean of 10 model prediction repeats, shaded area indicates the standard deviation, solid lines show observed results in HTS data. (**D**) As C., but as a function of the slope of the power-law distribution. (**E**) Graphical representation of parameter sweep results for prediction of CD4⁺ and CD8⁺ repertoires from αβ clone-size distributions following a mixture model consisting of singleton clones and a small fraction of large clones. The color represents goodness of fit, with dark green being better predictions for number of sequences per incidence in samples. Empty circles indicate parameter combinations resulting in qualitatively correctly predicted $\mathcal{P}(\sigma)$, that is 3 > 2 > 1 for TCRα and 2 > 1 for TCRβ and 2 > 3 for TCRβ. Filled circles indicate parameter combinations with the smallest distance to the incidence data and a correct $\mathcal{P}(\sigma)$ prediction.

The online version of this article includes the following figure supplement(s) for figure 4:

**Figure supplement 1.** Prediction of power-law model (exponent 2.3) for single sample data.

**Figure supplement 2.** Similar to *Figure 4*, but for HTS data from which TCRα and TCRβ sequences were removed that also occurred in the corresponding memory samples.

**Figure supplement 3.** Similar to *Figure 4*, but for HTS data processed with RTCR.

below a given abundance threshold. Therefore we explored whether a more generalised model, which does not make any assumptions about the distribution of the low abundance T cells, would predict our experimental data as well as the power-law model. In this simple mixture model we generate a population in which the majority of cells are present only once, and a minority are present many times. We scanned the parameter space of this model, varying both the proportion of cells in each population, and the size of clones in the larger population. The prediction of the model for each parameter pair was compared to the experimental data from experiment 2, both for the number of TCRs (combining α and β sequences) observed in one, two or three subsamples, and for the median $\mathcal{P}(\sigma)$ of these TCR sequences. The best agreement between model and data was observed when 1–5% of the cells were derived from abundant T-cell clones (between $10^5$ and $10^6$ cells in the whole repertoire) (*Figure 4E*).

## Abundant T-cell sequences are enriched for zero insertions and for antigen-association

In human prenatal thymocytes, the enzyme terminal deoxynucleotidyl transferase (TdT) is not expressed, leading to the production of TCR sequences with zero insertions of N-nucleotides. Pogorelyy and colleagues showed that enrichment of zero insertion TCR sequences can be used to detect fetal clones even in adults, and that their contribution to the overall repertoire decays slowly with age (*Pogorelyy et al., 2017*). Interestingly, the proportion of zero-insertion sequences was strongly enhanced in those sequences observed more than once in the three subsamples examined in experiment 2 (*Figure 5A*). The interpretation of this finding is not straightforward, since zero-insertion TCR sequences have higher median generation probabilities, and this is also a property of abundant sequences as discussed above. Nevertheless, the data are compatible with a model in which the large clones observed in the repertoire are generated preferentially during early prenatal development of the naive T-cell repertoire.

We next examined if the abundant sequences in our data showed characteristics of semi-invariant NKT and MAIT cell populations. Classical NKT cells are characterized by an invariant TRAV24-TRAJ18 α chain and β chains with TRBV11 (*Dellabona et al., 1994*). MAIT cells are enriched for TCRα rearrangements of TRAV1-2 with TRAJ33, TRAJ12 and TRAJ20 (*Reantragoon et al., 2013*), and TCRβ sequences predominantly using TRBV20 and TRBV6 (*Lepore et al., 2014*). Since our HTS data does not contain information on αβ pairing, we studied both chains separately. A substantial fraction of the observed TCRβ sequences matches the characteristics of MAIT cells, and to a lesser extent NKT cells (*Figure 5B and C*). For both cell types, however, this fraction does not show a clear relation to incidence, and does not suggest enrichment for MAIT or NKT cells among abundant sequences. The most abundant TCRα sequences are enriched for NKT sequences, but these still account for only a small fraction of the total (0.3% and 1.7% for CD4⁺ and CD8⁺, respectively, *Figure 5B*). Hence, we conclude that only a small fraction of the abundant sequences are derived from clones with a MAIT or NKT cell phenotype.

Finally we analysed whether the abundant TCR sequences in the naive population could be detected in a database of TCR sequences with known antigen specificity (*Shugay et al., 2018*). Interestingly, there was a striking enrichment of TCR sequences with known antigen-specific annotation within the high abundance TCRα sequences observed in more than one subsample from experiment 2, and to a lesser extent for TCRβ sequences (*Figure 5C*). Interpretation is again not straightforward, because the high generation probabilities of the abundantly observed chains could lead to these sequences being over-represented in the database (ascertainment bias). Additionally, the observation may also reflect the fact that the naive T-cell populations we sequenced contained some antigen-experienced T cells with a naive phenotype (*Pulko et al., 2016*). Finally, the observation is also compatible with the hypothesis that TCR recombination has evolved to preferentially generate TCRs specific to common pathogens like CMV or EBV as discussed in *Thomas and Crawford (2019)*.

## Discussion

The diversity and clone size distribution of the naive T-cell repertoire has been the subject of considerable debate, fueled by the difficulty of obtaining more than a very small sample of the total repertoire, and by a variety of other technical considerations which we address in this study. We use a quantitative UMI-based sequencing protocol, and careful error correction to analyse the naive and

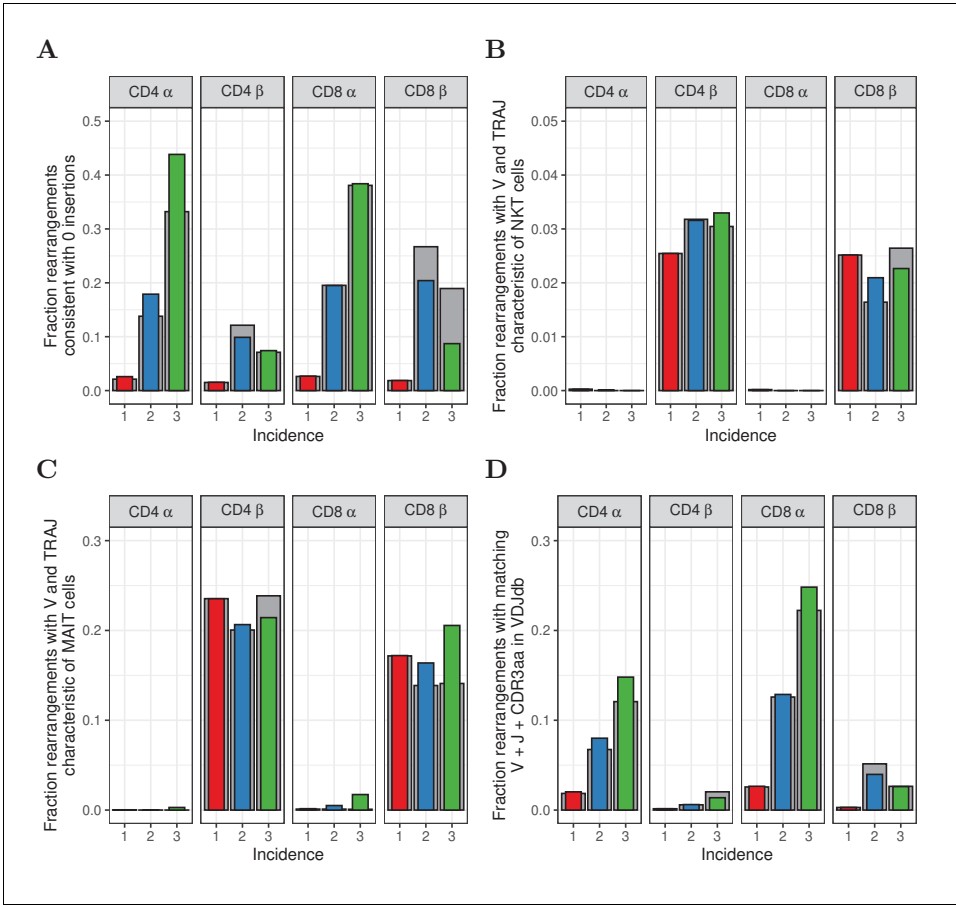

**Figure 5.** Characterization of abundant TCRα and TCRβ sequences. (A) The fraction of rearrangements with zero minimal N-additions for sequences observed in 1, 2 or 3 naive subsamples. Data are shown without (colored bars) and with cleaning of overlap with memory (grey bars). (B) Fraction of TCRα and TCRβ sequences with V(J) usage characteristic of NKT cells (TRAV24-TRAJ18 for TCRα; TRBV11 for TCRβ). (C) Fraction of TCRα and TCRβ sequences with V(J) usage characteristic of MAIT cells (TRAV1-2 with TRAJ33, TRAJ12 or TRAJ20 for TCRα; TRBV20 or TRBV6 for TCRβ). (D) Fraction of sequences having at least one match (CDR3 amino acid sequence as well as V and J annotation) with the VDJdb (*Shugay et al., 2018*).

The online version of this article includes the following figure supplement(s) for figure 5:

**Figure supplement 1.** Similar to *Figure 5*, but for HTS data processed with RTCR.

memory repertoires from three healthy human volunteers. We convincingly demonstrate that a small proportion of the TCR sequences are present more than once in a sample of naive T cells from blood, corresponding to expected frequencies greater than 1 in $10^5$. This number of abundant TCRα sequences is higher than the number of abundant TCRβ sequences.

We carefully considered different mechanisms that could give rise to these abundant TCR sequences. We examined the contribution of potential contamination of the naive population with abundant T cells from the memory compartment during the sorting process, but the extent of such contamination was small (for CD8$^+$ cells) or not detectable (for CD4$^+$ cells). Furthermore, exclusion of all TCR sequences which occurred in both memory and naive populations did not alter the subsequent conclusions of the analysis. We also considered the possibility that abundant TCR sequences were observed due to sampling multiple mRNA molecules from the same cell. In order to exclude this possibility, we carried out an experiment where we divided up a sample of sorted naive T cells into three subsamples prior to lysis, and sequencing. In this experimental paradigm, TCR sequences found in more than one subsample must arise from different T cells. We observed that repeat sampling of mRNA from the same T cell did indeed occur, and might account for as much as 75% of the high abundance TCRβ sequences (for which there are more mRNA molecules per cell, [*Oakes et al.,*

2017]), and as much as 25% of the TCRα sequences. However, this effect was mostly restricted to TCR sequences observed twice, and made little contribution to TCR sequences observed three or more times.

Having excluded methodological causes of high abundance TCR sequences, we examined two biological mechanisms which could explain the data. The first mechanism we consider is that abundant sequences derive from identical TCRα and TCRβ rearrangements occurring in multiple clones. In this model, abundance arises not from multiple sampling from the same large clone of T cells, but from summation over many different clones of T cells, each of which share an α or β chain. The second mechanism is that the naive repertoire clone size distribution is not uniform, but contains many small and some large clones. We combine computational models with experimental data to provide evidence that both mechanisms are required to explain the observed data. The first mechanism dominates the repertoire of TCRα, and is likely to contribute to the majority of observed abundant sequences. Interestingly, the model suggests that those TCRα which have the highest probability of generation are produced hundreds of thousands, or even millions of times within an individual, and must therefore be produced extremely frequently in the thymus. In contrast, the first mechanism has a smaller impact on the TCRβ repertoire, and abundant TCRβ sequences are more likely to arise from large clones in the naive repertoire.

The experimental limitations of sampling small volume of blood which contains only a tiny proportion of the total repertoire has dramatic effects on the observed TCR frequency distribution. One can use the analytical solution of the neutral model (Section 'Neutral model for dynamics of naive T cells') with thymic introduction size $c = 1$ to illustrate this extreme sampling effect: $\hat{F}_i \approx F_i(\frac{s}{\theta})^i$, where $\hat{F}_i$ and $F_i$ are the number of clones present with $i$ cells in the sample, and in the pool, respectively, and $s$ is the fraction of the repertoire that was sampled (here s ~ $10^{-6}$). Since $s/\theta$ is of order $10^{-5}$ and this is raised to the $i$th power, even very large TCR clones become rare in such a sample. Because of this, it is difficult to be definitive about the exact underlying T-cell clone frequency distribution which gives rise to the abundant TCR sequences we observe. The data are certainly compatible with a power-law distribution, as has been suggested previously (*Desponds et al., 2016*). But many distributions made up of a mixture of rare clones and a small proportion (1–5%) of large clones ($10^5$ - $10^6$) are compatible with the data we observe.

The demonstration of large clones in the naive repertoire raises the question of what determines the different sizes of different clonotypes. The neutral model already excludes repeated thymic production as explanation for large clones, because the combined probability of repeated αβ-clone production is very low (*Dupic et al., 2019*). We confirmed that the abundant TCR sequences were not strongly enriched for sequences characteristic of iNKT and MAIT cells (*Figure 5B&C*). An alternative explanation is that the large clones may actually be antigen-experienced, but with a naive phenotype such as memory stem T cells (*Gattinoni et al., 2011*; *Lugli et al., 2013a*; *Lugli et al., 2013b*; *Fuertes Marraco et al., 2015*; *Pulko et al., 2016*). However, a number of alternative explanations for this enrichment exist, as discussed above. Furthermore, antigen-experienced T cells should be present in the memory populations, and large clones could still be observed even after removal of all cells which occur in both memory and naive populations. So, although we cannot exclude that some of the frequent TCR sequences may be derived from T cells that are not truly naive, we believe the data argue for the existence of truly naive large clones.

We speculate that the most likely mechanism for large clones is preferential growth/survival of some clones, presumably due to preferential selection on self-peptide/MHC (*Rudd et al., 2011*; *Lythe et al., 2016*). Intriguingly, the abundant TCR sequences we observed were enriched for sequences without N-insertions, a characteristic of TCRs produced prenatally (*Pogorelyy et al., 2017*). The large clones may therefore be established very early in the development of T-cell adaptive immunity, before homeostasis of the immune system is achieved and when more rapid division and clonal expansion may be favoured.

In conclusion, our study highlights the huge impact of subsampling on correct interpretation of TCR repertoire data. It provides evidence for two different mechanisms which give rise to abundant TCR sequences in the naive human repertoire. The first mechanism, driven by multiple identical recombination events, is frequently overlooked in the analysis of T-cell repertoires, but has important implications in interpretation of observed sharing between different T-cell subpopulations of an individual, and between individuals (public TCR sequences). The second mechanism suggests that the

TCR sequence plays a critical role in naive T-cell homeostasis. Further experiments will be required to fully elucidate the cellular and molecular mechanisms which underlie the heterogeneity of the naive T-cell repertoire.

## Materials and methods

### Cell sorting and sequencing

Sequence reads came from T cells extracted from blood samples of three healthy volunteers, between 30 and 40 years old. Using CD27 and CD45RA markers, FACS-sorting was performed, identifying naive (CD27$^+$CD45RA$^+$), CM (central memory, CD27$^+$CD45RA$^-$), EM (effector memory, CD27$^-$CD45RA$^-$) and EMRA (effector memory RA, CD27$^-$CD45RA$^+$) cells. Barcoded TCRα and TCRβ cDNA libraries were obtained by reverse transcription of RNA molecules coding for either the α or β chain, respectively, followed by single strand DNA ligation to attach unique molecular identifiers (UMIs) of 12 nucleotides. These were PCR-amplified and sequenced using the Illumina MiSeq platform. For full description of the sequencing procedure, we refer to *Oakes et al. (2017)* and *Uddin et al., 2019*. The raw sequence files are available on the Sequence Read Archive (https://www.ncbi.nlm.nih.gov/sra, RRID:SCR_004891) as experiment SRP109035.

### Sequence analysis

We used the Decombinator pipeline (*Thomas et al., 2013*) (Version 3.1, RRID:SCR_006732) to demultiplex, annotate, and error-correct the raw sequencing reads. Our reads contain UMIs of 12 base pairs that can be used to identify which TCRα or TCRβ sequences are derived from the same cDNA molecule. Decombinator performs error correction on sequences by collapsing those that are similar and are associated with the same UMI. The pipeline also error corrects UMIs, collapsing those UMIs that are associated with the same TCRα or TCRβ sequence and differ from each other by 2 or fewer sequence edits (i.e. the default barcode threshold). This error correction assumes it is unlikely for any sequence, irrespective of its frequency, to contain two UMIs that are nearly identical, concluding the UMIs are different because of PCR or sequencing errors.

We improved this by setting the barcode threshold to 0 and replacing it by an UMI error correction algorithm that takes the number of UMIs into account. Consider a TCRα or TCRβ sequence supported by $i$ different UMIs, that is with frequency $i$. The Hamming distance, $H$, between two random UMIs of 12 base pairs can be represented by a binomial random variable, $H \sim B(n,p)$, where $n = 12$ and $p = \frac{3}{4}$ (assuming uniform frequencies of the 4 different bases). There are $\binom{i}{2}$ distinct comparisons between the $i$ UMIs, and assuming that every comparison is independent, the expected distribution of Hamming distances is $n_i(h) = \binom{i}{2} \mathcal{P}(H = h)$. To determine whether two UMIs are unexpectedly similar, we define a threshold distance that depends on the frequency of their TCRα or TCRβ sequence ($i$):

$$D_\alpha = \max(\{d : \sum_{h=1}^{d} n_i(h) \leq \alpha\}) . \tag{1}$$

Our algorithm corrects UMIs for a given sequence as follows: From $d = 1$ to $d = D_\alpha$, for all UMI pairs with $H \leq d$, add the read count of the less frequent UMI to the more frequent UMI and remove the former. We applied this algorithm to every TCRα and TCRβ sequence in our HTS data using $\alpha = 0.05$. The effects of this correction method are shown in *Figure 6*. After the improved correction, the distribution of Hamming Distances within and between distinct TCRα and TCRβ sequences is very similar, indicating that most erroneous UMIs have been removed. Our improved correction decreases the estimated frequency of many sequences at low frequencies, which indicates that many TCRα and TCRβ sequences that were observed two or three times, are actually singletons for which the UMI was mutated once or a few times. In the example given in *Figure 6*, the number of sequences that were observed more than once decreased with 66% by our improved correction (from 11491 to 3855), whereas the default correction estimated 9342 (only 19% reduction) of the sequences to have more than one true UMI.

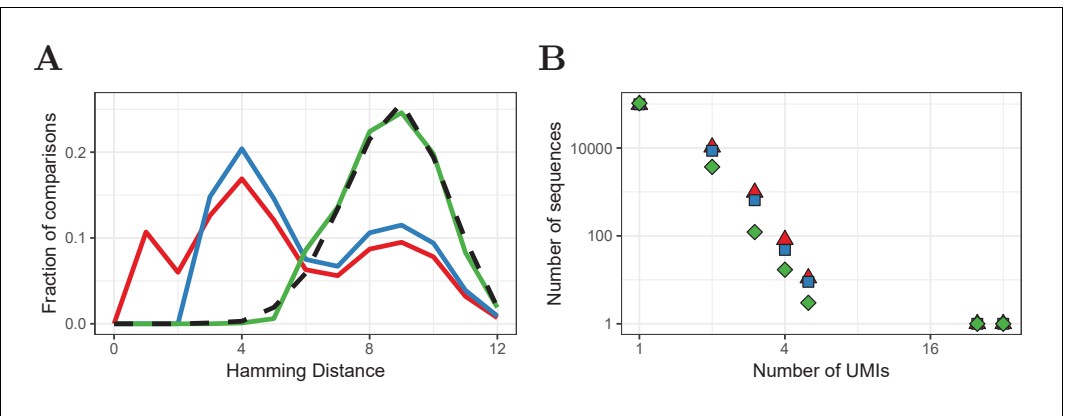

**Figure 6.** Improved UMI correction leads to more reliable estimation of sequence frequencies. (**A**) Distribution of Hamming Distances of UMIs within TCRβ sequences (naive CD4[+] sample of volunteer 1) before correction (red), after default correction (blue) and after improved correction (green), in comparison with the distribution of UMIs between sequences (black dashed). (**B**) Distributions of the same TCRβ sequences after the different correction strategies. Frequently observed TCRβ sequences remain at the same frequency after correction, whereas the frequency of other sequences tends to be overestimated due to mutated UMIs, which is compensated for by improved UMI correction.

Because our analysis focuses on the naive T-cell repertoire, we combined the different memory populations by adding the abundance of identical TCR sequences (V and J annotation as well as CDR3 nucleotide sequence) in the corresponding CM, EM and EMRA samples. We included for analysis the sequences that were reported as functional by Decombinator and had non-zero $\mathcal{P}(\sigma)$. We also processed the HTS reads with RTCR (*Gerritsen et al., 2016*) (Version 0.4.3). This pipeline determines a sample-based error rate and uses this rate to perform clustering on reads. Compared to Decombinator, RTCR estimates our reads to contain more PCR and sequencing errors and therefore tends to be more conservative in terms of reported diversity. Because RTCR reports fewer distinct rearrangements per sample, the overlap between samples (i.e., the number of chains with incidence 2 and 3) is lower than in Decombinator output. For each of the main-text figures, a supplemental RTCR-based version is provided. Although the quantitative results are not identical, the RTCR results qualitatively match those of the Decombinator output, confirming that our results are not algorithm-dependent.

## Subsampling to exclude inflated abundance through multiple RNA contributions by single cells

An important step in our analysis is the additional experiment in which the naive cells were split into three parts before mRNA extraction. The probability for a naive cell to be sampled from the pool is very low ($< 10^{-5}$), but once a cell has been sampled it may likely contribute multiple RNA molecules. These would then be sequenced with different UMIs, inflating the abundance we measure in a sample. Hence, we use subsampling to avoid the noise on TCRα and TCRβ abundance introduced by variable TCR expression between cells. To quantify the possible effect of single cells contributing multiple RNA molecules, we performed a permutation test. We computationally joined the sequences observed in the three independently sequenced replicates, adding the abundance (as measured by UMIs) in each of the three subsamples together. We then randomly assigned the UMIs of these sequences to one of three artificial portions and again scored the incidence of all TCRα and TCRβ sequences. In this setting, RNAs contributed by single cells in a single sample, can be distributed over multiple permuted samples. This was done 10 times for each set of sequences and we found that permutation led to a large increase in the number of sequences occurring in multiple samples (*Figure 2—figure supplement 1A*). We quantified the number of abundant chains, by counting sequences observed in multiple samples (*Supplementary file 2*).

When multiple RNA molecules from a single cell can contribute a UMI (i.e., in the permuted set and within a single sample), the number of abundant sequences is greatly overestimated. About

25% of abundant α chains in this setting is actually due to inflated counts. For β chains the effect is much larger, with over 75% of abundance due to RNA content. This difference is consistent with our previous finding that T cells contain in the order of 300 *TCRB* and 100 *TCRA* RNA molecules per cell (*Oakes et al., 2017*). Moreover, the lower $\mathcal{P}(\sigma)$ values of β chains readily explains that there are fewer true duplets and triplets than for α chains. Subsampling appears to be very important when obtaining our most surprising result that high $\mathcal{P}(\sigma)$ values are enriched for β chains with incidence 2, but not incidence 3. After permutation, most duplets are due to RNA content (*Supplementary file 2*) and therefore no longer enriched for high $\mathcal{P}(\sigma)$ (*Figure 2—figure supplement 1B*). These results highlight the importance of our additional step of taking a single blood sample, dividing it into three portions and then analyzing all three subsamples separately.

## Sharing of TCRα and TCRβ sequences

We sequenced TCRα and TCRβ from whole blood samples taken from 28 healthy volunteers. The study was carried out in accordance with the recommendations of the UK Research Ethics Committee with written informed consent of all subjects. All subjects gave written informed consent in accordance with the Declaration of Helsinki. The protocol was approved by the University College London Hospital Ethics Committee 06/Q0502/92. The raw sequence files are available on the Sequence Read Archive (https://www.ncbi.nlm.nih.gov/sra, RRID:SCR_004891) as experiments SRP045430 and SRP151125. In order to measure how public the individual sets of sequences were, we measured their degree of sharing between our naive samples and these whole blood repertoires.

As shown in *Figure 2*, we have three sets of sequences, those with incidence 1, 2 and 3. For each set, we measured which fraction is also found in the 28 independent whole blood samples, which delivers 28 estimates of sharing. More precisely, we counted the number of shared TCRα and TCRβ sequences between the sets of sequences observed in two and three naive subsamples, and compared these to sharing with an equal size sample of naive sequences which were only observed in one subsample. Since the number of sequences which occurred more than once was much smaller than the number of sequences which only occurred once, we subsampled the set of unique sequences 10 times. The results are shown as the number of shared TCRα or TCRβ for each whole blood repertoire, as a proportion of their number of sequences in the samples being tested (*Figure 2—figure supplement 3A*). In order to study the sharing of the β chains in our data with higher resolution, we also analyzed overlap of the sets of sequences with the TCRβ data from a large cohort of 786 people published in *Emerson et al. (2017)*; *Figure 2—figure supplement 3B*.

## Neutral model for dynamics of naive T cells

To model naive T-cell dynamics in the absence of peripheral selection, we developed a model that is similar to the Neutral Community Model (NCM) of *Hubbell (2001)*. Naive T cells, viewed through an ecological lens, are individuals, and all naive T cells sharing the same TCRα and TCRβ sequence are part of the same species (αβ-clone). Neutrality, as defined by Hubbell, means that all species have the same per capita probability of birth (peripheral division) and death. When considering the model, we ignore the very small chance that an existing αβ-clone is produced again by the thymus. Hence, in our simulations we assume that the thymus produces T-cell clones that are unique and novel.

Consider a pool of $N$ naive T cells belonging to clones, each consisting of $i$ cells, which changes by thymic production, cell division and cells leaving the naive pool (as a result of cell death or activation). During each event, one randomly selected cell exits the pool, causing the corresponding clone to decrease in size from $i$ to $i-1$ cells. With probability $1-\theta$, another randomly selected cell will divide, causing the corresponding clone to increase its size from $i$ to $i+1$ cells. Alternatively, with probability $\theta$, thymic production *can* occur: every $c$ events in which no peripheral division occurred, the thymus will release $c$ cells of a newly produced clone. So, the pool size $N$ only fluctuates by $c$ cells, and because $N \gg c$, the total number of cells stays almost constant during the entire simulation. The per capita birth rate $((1-\theta)/N)$ and death rate $(1/N)$ are equal for all T-cell clones, which makes this a neutral model. In this discrete-time model, exit and production are coupled, but its dynamics can be approximated by a continuous-time model, in which thymic production, cell division, and deaths are uncoupled Poisson processes. This is illustrated by the Markov

chain depicted in *Figure 7*, in which the states are clone sizes and the rates show the probabilities of clones moving to another state.

This Markov process describes the dynamics of the clone-size distribution $F$, that is the total number of clones $F_i$ consisting of $i$ cells. After many birth and death events, individual clones still change in clone size over time, but the clone-size distribution approaches equilibrium. At this steady state, the rate at which new clones enter the naive pool, $\theta/c$, equals the rate at which clones leave the pool, that is $F_1(1/N)$. Hence, in equilibrium, the number of singletons, clones with only one cell, approaches $F_1 = \theta N/c$. The total rate at which the cells of clones with $i$ cells divide and die depends on the total number of cells belonging to $F_i$ clones: $iF_i$. For clone sizes up to $c$ cells, the rate at which the cells of the $F_i$ clones die, ($iF_i/N$), balances the division the cells of $F_{i-1}$ clones $((i-1)F_{i-1}(1-\theta)/N)$ and the rate at which new clones enter the pool ($\theta/c$). The analytical solution to this recurrence relation $iF_i/N = (i-1)F_{i-1}(1-\theta)/N + \theta/c$ is:

$$F_i = \frac{N - N(1-\theta)^i}{ic}, \quad \text{for } 1 \leq i \leq c . \tag{2}$$

For states with $i>c$, only birth and death of cells need to balance between states $i-1$ and $i$ (as there is no net flux from clones introduced by the thymus): $iF_i/N = (i-1)F_{i-1}(1-\theta)/N$. This recurrence relation has the following analytical solution:

$$F_i = \frac{cF_c(1-\theta)^{i-c}}{i}, \quad \text{for } c \leq i \leq N . \tag{3}$$

When predicting the full clone-size distribution, we use *Equations 2 and 3* to calculate the steady-state distribution. The total number of all distinct clones (i.e. the richness) in the steady-state repertoire is simply the sum over all their frequencies $F_i$, $R = \sum_{i=1}^{\infty} F_i$, which has a simple closed-form solution for $c=1$,

$$R = \sum_{i=1}^{\infty} F_i = \frac{\theta N \ln \theta}{\theta - 1} \quad \text{for } c = 1 . \tag{4}$$

The Simpson's diversity of the steady state repertoire also has a simple form,

$$S = 1 / \sum_{i=1}^{\infty} F_i \left(\frac{i}{N}\right)^2 = \frac{2\theta N}{2 + (c-1)\theta} , \tag{5}$$

which equals $F_1 = \theta N$ for $c=1$, and is a saturated function of $\theta$ if $c>1$.

We consider the sampling process of a small fraction $s$ from a naive T-cell pool of $N$ cells, which clones follow the distribution $F$ in *Equation 2* and *Equation 3*. Assuming the naive pool is large and well-mixed, the number of T cells, $X$, sampled from the $j$ cells belonging to a particular clone, can be approximately represented by a binomial random variable, $X_j \sim B(n = j, p = s)$. The expected clone-size distribution of the sample, $\hat{F}$, is then given by

$$\hat{F}_i = \sum_{j=i}^{N} F_j \mathcal{P}(X_j = i) . \tag{6}$$

The strong distortion of sampling from clone-size distributions can be illustrated using the analytical solution of *Equation 6* for the neutral model for $c=1$:

$$\hat{F}_i = F_i \left(\frac{s}{s + (1-s)\theta}\right)^i . \tag{7}$$

Since $s$ is typically very small, this equation can be simplified to $\hat{F}_i \approx F_i(\frac{s}{\theta})^i$ (as $s \ll \theta$), which clearly shows that even very abundant clones will become rare or absent in a small sample.

## Clone-size distributions of the naive T-cell pools

Since our data contains separate data on both CD4[+] and CD8[+] T cells, we predicted the clone-size distributions of both subsets separately. To account for the larger CD4[+] pool (*Wertheimer et al.,*

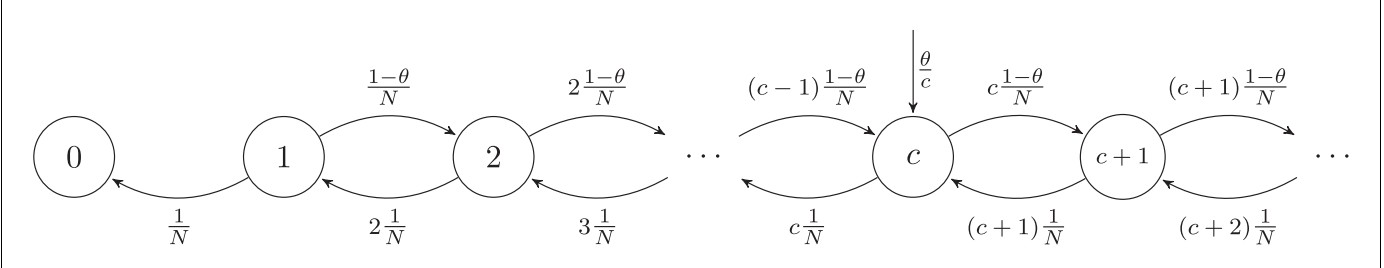

**Figure 7.** Markov chain representation of the neutral model with thymic introduction size $c$.

2014; *Westera et al., 2015*), we set its pool size $N = 7.5 \times 10^{10}$ cells, while we used $N = 2.5 \times 10^{10}$ for the naive CD8$^+$ pool.

When analyzing the neutral model, we used its steady-state distribution (*Equation 2* and *Equation 3*). Since the β chain rearranges first, followed by a few divisions before rearrangement of the α chain (*Gonçalves et al., 2017*), we use $c = 100$ for TCRβ and $c = 10$ for TCRα. We also used various phenomenological clone-size distributions that are not based on a mechanistic model. To allow for exploration of a wide range of distributions, we chose mathematical functions which form can be changed by a single parameter, such as the slope of the power-law distribution.

The power-law distribution with form $F_i = F_1 \times i^{-k}$ shows a straight line on a log-log plot. Since all $F_i$ are written as a function of $F_1$, the total number of cells $N = F_1(1 + 2 \times 2^{-k} + 3 \times 3^{-k} + ...) = F_1 \sum_{i=1}^{\infty} i^{1-k}$. This sum is convergent for $k>2$ and gives

$$F_i = \frac{N i^{-k}}{\zeta(k-1)}, \quad \text{for } k>2 \tag{8}$$

for the power-law clone-size distribution, in which $\zeta$ is the Riemann zeta function.

We also studied repertoires with log-normal distributions of clone-sizes by drawing from a normal distribution and raising 10 to the power of these numbers for clone sizes. For this we used varying μ and $\sigma = \mu/10$. These distributions yielded results that were qualitatively similar to those from the neutral model (not shown). For the simple mixture model (*Figure 4E*), we defined two populations of clones: (1) singletons (clones of just one cell that can only contribute to high TCRα or TCRβ abundances by sharing a chain with many other clones) and (2) large clones of equal size. We varied the fractions of both populations as well as the size of the large clones to find which fraction of the cells in the naive repertoire is expected to belong to large clones. A similar analysis, combining the aforementioned distribution following from the neutral model with a log-normal distribution for the population of large clones, produced very similar results (not shown).

## In silico samples from modelled clone-size distributions

To compare the clone-size distributions with the HTS data of the blood samples, we generated TCRα and TCRβ repertoires using IGoR (*Marcou et al., 2018*). We generated $10^8$ TCRα and TCRβ sequences using IGoR's default recombination model and parameters. We selected the rearrangements which CDR3 nucleotide sequence consisted of a multiple of 3 nucleotides (in frame) and did not contain in-frame stop codons, in line with the inclusion criteria of productive rearrangements in our HTS samples (~28%). Next, we calculated generation probabilities $\mathcal{P}(\sigma)$ for all these rearrangements. This may seem a detour, but this is needed as many different scenarios can lead to the same TCRα or TCRβ rearrangement.

Only a small percentage of thymocytes that undergo rearrangements in the thymus will eventually be exported as a naive T cell. This is due to out-of-frame rearrangements, but also as a result of both positive and negative selection. Moreover, the generation probability distributions of pre- and post-selection TCRα and TCRβ repertoires are markedly different (*Elhanati et al., 2014*). To account for these observations, we train a $\mathcal{P}(\sigma)$-dependent selection model to account for the effects of thymic selection on our IGoR-produced TCRα and TCRβ sequences. Note that this selection method is

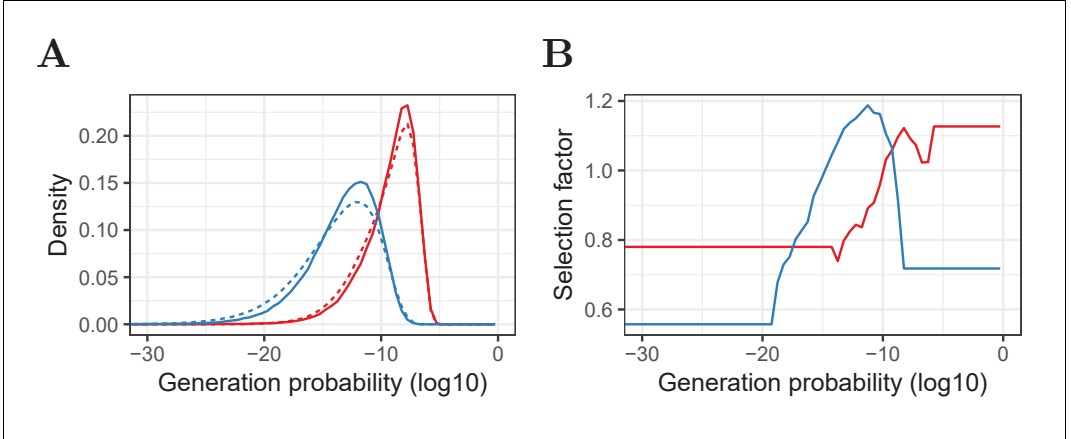

**Figure 8.** Pre- and post-selection $\mathcal{P}(\sigma)$ densities and $\mathcal{P}(\sigma)$-dependent selection factors for α and β chains. (**A**) Relative frequency of generation probabilities of TCRα (red) and TCRβ (blue) sequences in the combined HTS data (solid) and IGoR output (dashed). (**B**) The bin-specific selection factors $f_{\mathcal{P}(\sigma)}$ are determined by division of the density of a given bin in the HTS data by the density in the pre-selection IGoR output. A value of 1 means that a sequence with this $\mathcal{P}(\sigma)$ has an average probability to be selected in the thymus, whereas lower values indicate stronger selection and higher values weaker selection (i.e., a higher probability to pass selection).

based on single chains rather than on αβ-TCRs. This is because recombination of β and α chains occurs at different points in T-cell differentiation. The first step in selection, after formation of the β chain, is based on correct folding and expression, using a pre-α pseudochain for pairing. If the T cell survives this step, it undergoes multiple rounds of divisions, by which its β chain can pair with many different α chains. The second step is positive and negative selection based on MHC-peptide interactions, which is likely to operate on a joint αβ pair. It is unknown how much each of these two steps contributes to the overall selection process.

For TCRβ selection, we reason that selection on pairing with the invariant pre-α chain acts exclusively on the level of single β chains, and once the T cell survives this first step, it is expected to survive with at least one of the many α chains it can pair with during the second step. The absence of strong structural constraints on αβ pairing supports this idea (*Tanno et al., 2020*). Additionally, the large $\mathcal{P}(\sigma)$ shift between pre- and post-selection TCRβ repertoires is indicative of selection acting on the level of single β chains (i.e., the probability for a β chain to be selected is largely irrespective of the α chain). For α chains this shift is less pronounced, and a newly generated α chain only pairs with a single β chain. We therefore also tested the effect of an alternative selection model in which a given α chain survives selection with a given probability for repeated production events, reflecting different selection outcomes when pairing with different β chains. This approach decreases the average frequency of α chains in the post-selection repertoire (since in this case they will on average survive only in a fraction of selection events, instead of our default all-or-nothing model). This did not affect our results in a qualitative manner and we proceeded with selection on the level of single chains for both TCRα and TCRβ.

We use each of the HTS data sets from the single sample experiment (shown in *Figure 1*) to calculate the relative enrichment or depletion of 100 log10 $\mathcal{P}(\sigma)$ bins (ranging from −50 to 0) compared to 100 equally sized samples of the IGoR output, for TCRα and TCRβ separately. If the HTS data contained few rearrangements for a given bin, we joined adjacent bins in such a way that the bin-specific selection factor was always based on at least 1% of the experimental observations (*Figure 8*). This approach yielded $\mathcal{P}(\sigma)$-specific selection factors $f_{\mathcal{P}(\sigma)}$ ranging from 0.6 to 1.15 (i.e., our data suggests that sequences with a preferable $\mathcal{P}(\sigma)$ are about 2 times as likely to be selected as those in the least preferable $\mathcal{P}(\sigma)$ domain). We assumed an overall selection factor of 1/3, meaning that one out of 3 productive TCRα or TCRβ rearrangements would survive selection. We then allowed sequences to be part of the post-selection repertoire with probability

$$p_{selected} = f_{\mathcal{P}(\sigma)}/3 \tag{9}$$

and stored the outcome to make a consistent decision when multiple copies of the same TCRα or TCRβ sequence were present in the pre-selection repertoire. This approach yielded post-selection repertoires with $\mathcal{P}(\sigma)$ distributions similar to the single sample HTS data. Other values for the overall selection probability, ranging from 1/10 to 1, were also tested, but yielded similar qualitative results (not shown).

We could have assigned all clones in the clone-size distribution an α and β chain with this approach. However, since only a very small part of the repertoire is sampled, we chose to only assign an identity to those clones present in the samples. Hence, we started with predicting the presence of all clones, as a function of their size, in each of the samples. The probability that a clone with $i$ cells is represented by at least one cell in a sample of $n$ cells from a pool of $N$ cells is

$$p_i = 1 - (1 - \frac{i}{N})^n \tag{10}$$

Given $F_i$, which is the number of clones in the pool with clone size $i$, the number of these clones present in the sample of $n$ cells can be approximately represented by a binomial random variable, $X_i \sim B(n = F_i, p = p_i)$. We evaluate this for the entire clone-size distribution $F$. $N$ and $F$ are known from the model but one cannot directly determine the number of sampled cells $n$. This is because individual cells may contribute multiple mRNA molecules and many cells may have been present in the FACS-sorted sample without contributing mRNA to the eventual sequenced fraction. Therefore we learn the sample size by assigning α or β to sampled clones and choosing $n$ such that the predicted diversity (i.e., number of distinct chains) matches the experimental observations. We took the number of distinct TCRα or TCRβ sequences as lower bound for the sample size, since in this model individual cells are assumed to express one functional α or β chain. The total number of cells reported by the FACS-sorter was used as upper bound. We also checked the implications of the observation that some T cells contain two functional α and/or β chains, but this did not qualitatively change our results (not shown).

Thus, we adjusted the generation probability distribution by training a $\mathcal{P}(\sigma)$-dependent selection model on independent HTS data and based the sample size on the corresponding subsamples. Hence, the predicted individual subsamples reflect the experimental observations in terms of diversity and generation probabilities. We use the chains occurring in multiple samples (i.e., those with incidence 2 and 3) to assess the agreement between model predictions and the HTS data. We repeated the sampling process and assignment of α and β chains 10 times for each model-parameter combination to account for the stochastic nature of sampling and V(D)J recombination.

## Acknowledgements

We thank Laurens Krah for mathematical advice and helpful discussions. This work was supported by The Netherlands Organization for Scientific Research (NWO) Graduate Program 22.005.023 (to PCdG), the VIRGO consortium, which is funded by the Netherlands Genomics Initiative and by the Dutch government (FES0908) (to BG), by a grant to BC from Unilever PLC and supported by the National Institute for Health Research UCL Hospitals Biomedical Research. JH was supported by an MRC studentship.

## Additional information

### Funding

| Funder | Grant reference number | Author |
|---|---|---|
| Nederlandse Organisatie voor Wetenschappelijk Onderzoek | Graduate Program 22.005.023 | Peter C de Greef |
| Netherlands Genomics Initiative | FES0908 | Bram Gerritsen |

| Unilever | | Theres Oakes<br>Benjamin Chain |
| --- | --- | --- |
| National Institute for Health Research UCL Hospitals Bio-medical Research | | Benjamin Chain |
| Medical Research Council | Studentship | James M Heather |

The funders had no role in study design, data collection and interpretation, or the decision to submit the work for publication.

## Author contributions

Peter C de Greef, Conceptualization, Formal analysis, Visualization, Methodology, Writing - original draft, Writing - review and editing; Theres Oakes, Validation, Investigation, Writing - review and editing, Experimentation - T cell sorting and receptor sequencing; Bram Gerritsen, Conceptualization, Software, Formal analysis, Visualization, Methodology, Writing - original draft, Writing - review and editing; Mazlina Ismail, Software, Formal analysis; James M Heather, Software, Methodology, Writing - review and editing; Rutger Hermsen, Formal analysis, Writing - review and editing; Benjamin Chain, Rob J de Boer, Conceptualization, Supervision, Writing - review and editing

## Author ORCIDs

Peter C de Greef https://orcid.org/0000-0001-9802-629X
Rutger Hermsen http://orcid.org/0000-0003-4633-4877
Benjamin Chain https://orcid.org/0000-0002-7417-3970
Rob J de Boer https://orcid.org/0000-0002-2130-691X

## Ethics

Human subjects: The study was carried out in accordance with the recommendations of the UK Research Ethics Committee with written informed consent of all subjects. All subjects gave written informed consent in accordance with the Declaration of Helsinki. The protocol was approved by the University College London Hospital Ethics Committee 06/Q0502/92.

## Decision letter and Author response

Decision letter https://doi.org/10.7554/eLife.49900.sa1
Author response https://doi.org/10.7554/eLife.49900.sa2

# Additional files

## Supplementary files

• Supplementary file 1. Quantitative details about each dataset. Number of FACS-sorted cells in each sample, number of reads obtained for both $\alpha$ and $\beta$ and the number of UMIs that were accepted after Decombinator processing and UMI clustering. The naive subsamples were FACS-sorted and then divided into four equal aliquots. For the CD8$^+$ EMRA samples, TCR$\alpha$ and TCR$\beta$ were sequenced together for both volunteers in Experiment 1.

• Supplementary file 2. Fraction of TCR sequences with incidence > 1. For each dataset, the fraction of sequences observed in multiple subsamples is given, both in the data and after permuting the samples. From the fold-differences, the effect of single cells contributing multiple RNA molecules is estimated.

• Transparent reporting form

## Data availability

Sequencing data have been deposited in the Short Read Archive as experiments SRP109035, SRP045430 and SRP151125.

The following dataset was generated:

| Author(s) | Year | Dataset title | Dataset URL | Database and Identifier |
|---|---|---|---|---|
| Oakes T, Heather JM, Ismail M | 2018 | T cell receptor repertoire sequencing of healthy individuals | https://trace.ncbi.nlm.nih.gov/Traces/sra/?study=SRP151125 | SRP151125, NCBI Sequence Read Archive |

The following previously published datasets were used:

| Author(s) | Year | Dataset title | Dataset URL | Database and Identifier |
|---|---|---|---|---|
| Oakes T, Heather JM, Best K, Byng-Maddick R, Husovsky C, Ismail M, Joshi K, Maxwell G, Noursadeghi M, Riddell N, Ruehl T, Turner CT, Uddin I, Chain B | 2017 | TCR repertoire sequencing of T cell subsets from healthy individuals | https://trace.ncbi.nlm.nih.gov/Traces/sra/?study=SRP109035 | NCBI Sequence Read Archive, SRP109035 |
| Heather JM, Best K, Oakes T, Gray ER, Roe JK, Thomas N, Friedman N, Noursadeghi M, Chain B | 2014 | T-cell receptor repertoires in HIV-infected patients and healthy controls | https://trace.ncbi.nlm.nih.gov/Traces/sra/?study=SRP045430 | NCBI Sequence Read Archive, SRP045430 |
| Emerson R, DeWitt W, Vignali M, Gravley J, Hu J, Osborne E, Desmarais C, Klinger M, Carlson C, Hansen J, Rieder M, Robins H | 2017 | Immunosequencing identifies signatures of cytomegalovirus exposure history and HLA-mediated effects on the T-cell repertoire | https://doi.org/10.21417/B7001Z | immuneACCESS, 10.21417/B7001Z |

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
