## [Decision Letter]

**Acceptance summary:**

The paper addresses experimentally and theoretically the question of clone size distributions in T-cell repertoires. While we have known for a while that these distributions have long tails, experimental data has often not been extremely careful in correcting for biases and a careful measurement of different repertoire subsets is useful. Combining this with a model that tries to explain the origin of this distribution is an important element of the paper.

**Decision letter after peer review:**

Thank you for submitting your article "The naive T-cell receptor repertoire has an extremely broad distribution of clone sizes" for consideration by *eLife*. Your article has been reviewed by three peer reviewers, and the evaluation has been overseen by a Reviewing Editor and Satyajit Rath as the Senior Editor. The following individuals involved in review of your submission have agreed to reveal their identity: Mikhail V Pogorelyy (Reviewer #1).

The reviewers have discussed the reviews with one another and the Reviewing Editor has drafted this decision to help you prepare a revised submission.

All reviewers agree that the topic of the paper is interesting and important, however the consensus is that in its current form the paper will not appeal to a broad readership, since it is not very conclusive. At the same time the expert reader would be happier seeing all the biases clearly quantified and error bars clearly presented and discussed. As a result, the reviewers are calling for a really major revision (they provide concrete suggestions) and only if they are convinced the message of the paper is clear (e.g. predict observed bulk naive repertoire clone size distributions from a model, discuss differences subsets) will they recommend publication.

In general, the reviewers (the full reviews are attached for clarity on the details) make comments on presentation (citing previous work, putting this work in context for the general reader), being very quantitative and clear about biases and the experimental approach (see comments by reviewer 1) and (comments by reviewer 1 and 3) about the lack of conclusive modeling. Currently the reviewers feel the conclusion of the paper is "the naive T-cell receptor repertoire has an extremely broad distribution of clone sizes" but "we do not have a model that explains it". The reviewers understand that coming up with a completely conclusive model and ruling out all others is probably impossible. However, a more thorough theoretical discussion is needed, considering other models and arriving at one that fits the data (even if it may not be the only one that fits the data).

Finally, the reviewers make concrete suggestions to focus the manuscript on the most interesting aspects of this study worth developing further: a more detailed quantitative analysis of the CD4/CD8 and TCRα/TCRβ repertoires, the reported differences between TCRα and TCRβ, and the differing role of generation probability.

Reviewer #1:

The manuscript by de Greef et al. presents a computational framework to infer the clone size distribution of naive T cells. The main conclusion from this work is that there are large clones in the naive repertoire, the emergence of which could not be explained by the neutral model.

I think both methodology and data are interesting, but additional analysis is needed to address possible concerns outlined below.

1) The first dataset authors use consists of TCRA and TCRB repertoires of FACS-sorted naive and memory cells. As the authors note, FACS is not precise, and a result, naive subpopulation could be contaminated with abundant memory clonotypes. It would be useful to quantify the extent of this contamination. One way to visualize this is to make scatterplots of concentrations of each clonotype (both overlapping and non-overlapping) in naive vs. memory repertoire. As a control, such plots could also be done for naive and memory subsets of different individuals: there should be little correlation between clone frequencies and small overlap, since there should be no contamination on FACS in this case.

2) The end of the subsection “Abundant TCR sequences are frequently shared between naive and memory populations, and are enriched for high VDJ recombination probabilities”:

"if the overlap were the result of contamination only, the P(σ) of the [overlapping] sequences would be expected to reflect those of the memory subsets. Since the overlap is markedly enriched for high generation probabilities, most of it cannot be caused by contamination"

However, green dashed line (which is P(σ) of overlapping sequences) on Figure 1A for β chains (especially for CD8 in volunteer 2) seems to be very close to red lines (probs for memory clonotypes), suggesting that most overlap observed in β repertoires is caused by the contamination between two cell samples on FACS. I suggest modifying the text to avoid contradiction. Also, I think it would be useful to add theoretical prediction for the distribution of generative probabilities of overlapping sequences between mem and naive. If both memory and naive seqs have the same distribution of probs P(σ), then for overlapping(=recombined twice) sequences it should be P(σ) squared, if there is no contamination on FACS. Another way to obtain a prediction for the probability of overlapping sequences in the absence of contamination during FACS is to plot P(σ) for naive sequences of volunteer 1 overlapping with memory of volunteer 2 (and vice versa).

3) In the Discussion section, authors suggest that abundant clones in naive compartment could correspond to naive-like antigen-experienced subpopulation. It might be interesting to look for TCR amino acid sequences of these clones in existing databases of antigen-specific TCRs, such as VDJdb and McPAS.

4) Authors made additional experiment with splitting naive cells into three parts before the mRNA extraction to avoid the potential noise introduced by variance in TCR expression by different cells. Is it possible to quantify, how much bias is removed by using this design? E.g., what happens if we computationally join these three independently sequenced replicates back together, and then randomly assign each of UMIs to one of 3 portions? Are there many clones with elevated TCR transcription levels and thus inflated counts in bulk repertoires? Is this variance in TCR expression different for α and β chains? I think answers for these questions would be interesting for many groups sequencing TCR repertoires with RNA-based technology.

5) Authors fit parameters for different clone size distributions using this additional experiment with splitting naive repertoire into three parts. It would be interesting to compare resulting best fit prediction for clone size distributions to clone size distributions observed in bulk naive repertories of two volunteers, which are analyzed in the first part of the paper (e.g. rank-frequency scatterplots for model vs. data).

6) Another potential explanation for large naive clonotypes may be in the early repertoire development: it is known that TdT is not working, and thus first T-cells lack N-insertions. Previously our group have shown (see Figure 3 in Pogorelyy et al., 2017), that most abundant TCRβ from naive and cord blood (but not memory) repertoires are enriched with zero insertion clonotypes. It would be interesting to see, if abundant naive clonotypes observed in this study are also enriched with zero-insertion clonotypes (e.g. Figure 1B analog with some estimate of total number of N-insertions in each bin on y-axis).

Reviewer #2:

In this manuscript de Greef et al. study the clone size distribution in human naive T-cell receptor repertoires. They do it by a combination of data analysis of TCRA and TCRB sequences and computational modelling. After using FACS to sort the cells they utilized an UMI approach to sequence α and β chains of TCRs from naive and memory compartments of CD4 and CD8 cells. They first make a series of very interesting observations about the differences and commonalities between the generation probabilities (calculated by IGoR) in each cell type (CD4/CD8; naive/memory).

To explain these observations they adapt an ecological neutral model to T cell repertoires. By using this model they come to a surprising conclusion that the clone size distribution in naive T cells is not consistent with a neutral model and nor with a power law distribution. Instead they find that a subset of very large clones in the naive repertoire is essential to explain the data. They discuss an experimental potential artifact that could cause these observations and reason that this is likely not the case. Hence, their final conclusion and strong result is that naive T cell repertoires include a sub-population of very large clones. This conclusion opens a door to an undiscovered biology that determines the construction and dynamics of naive T cell repertoires.

I believe that this manuscript is of high interest to the whole immunology field.

Reviewer #3:

This work investigates potential determinants of the distributions of naïve CD4 and CD8 T cell clone sizes using computational analysis and mathematical modelling of high-throughput TCRα and TCRβ sequence data.

There are several major concerns about this manuscript:

1) Although this work improves on previous studies by considering both TCRα and TCRβ sequences in naive CD4 and CD8 T cell populations and using potentially more accurate quantification of clone sizes (using UMIs), the main conclusion of the manuscript, that the naïve TCR repertoire has a broad distribution of clone sizes, is not substantially novel. Heterogeneous clone sizes in naïve T cell repertoires have been reported, and investigated, in many previous studies (e.g. Robins et al., 2009, Quigley et al., 2010, Venturi et al., 2011, Qi et al., 2014, Pogorelyy et al., 2017). Previous studies have also reported substantial overlap of TCR sequences between naïve and memory CD8^+^ T cell compartments that is enriched for high abundance TCR sequences that have limited numbers of n-additions (e.g. Robins et al., 2010, Venturi et al., 2011). Furthermore, many studies have previously investigated the associations between V(D)J recombination / TCR production probability, naïve TCR clone size and inter-individual sharing of TCRs (e.g. Robins et al., 2009, Robins et al., 2010, Quigley et al., 2010, Venturi et al., 2011, Li PNAS 2014, Pogorelyy et al., 2017). Importantly, this manuscript has not acknowledged this substantial body of previously published and highly relevant research.

2) While the various models of the naïve T cell clone sizes may be novel, they did not provide sufficient new insights into the primary mechanisms driving the naïve T cell distribution. This was largely due to the fact that no one model considered by the authors could fully explain the observed TCR clone size distributions. One possible reason for this is the growing evidence for developmental- and age-linked heterogeneity in naïve T cell populations (e.g. Hogan et al., 2015, Rane et al., 2018, Reynaldi et al., 2019). Although the authors show model results for a range of parameters, this analysis does not account for the potential impact on the adult T cell repertoire of changes in naïve T cell dynamics over the lifespan of an individual. For example, it has been recently suggested that high abundance zero n-addition TCRs in the adult naïve repertoire have survived since early development and their high abundance is due to different homeostatic pressures in the peripheral repertoire during early development (Pogorelyy et al., 2017). This previous relevant research has not been considered in this manuscript.

3) A potentially interesting conclusion in this study is the limited association between TCRβ abundance, TCRβ production probability, and TCRβ sharing. This result is not consistent with the findings of many previous studies focused on TCRβ repertoires (listed above for concern #1). However, this discrepancy with previous studies was not discussed in the manuscript. Moreover, the authors have not undertaken further investigation to determine the robustness of this result to various parameters/assumptions in the computational analysis or potential explanations for this discrepancy.

---

## [Author Response]

The paper addresses experimentally and theoretically the question of clone size distributions in T-cell repertoires. While we have known for a while that these distributions have long tails, experimental data has often not been extremely careful in correcting for biases and a careful measurement of different repertoire subsets is useful. Combining this with a model that tries to explain the origin of this distribution is an important element of the paper.All reviewers agree that the topic of the paper is interesting and important, however the consensus is that in its current form the paper will not appeal to a broad readership, since it is not very conclusive. At the same time the expert reader would be happier seeing all the biases clearly quantified and error bars clearly presented and discussed. As a result, the reviewers are calling for a really major revision (they provide concrete suggestions) and only if they are convinced the message of the paper is clear (e.g. predict observed bulk naive repertoire clone size distributions from a model, discuss differences subsets) will they recommend publication.In general, the reviewers (the full reviews are attached for clarity on the details) make comments on presentation (citing previous work, putting this work in context for the general reader), being very quantitative and clear about biases and the experimental approach (see comments by Reviewer 1) and (comments by reviewer 1 and 3) about the lack of conclusive modeling. Currently the reviewers feel the conclusion of the paper is "the naive T-cell receptor repertoire has an extremely broad distribution of clone sizes" but "we do not have a model that explains it". The reviewers understand that coming up with a completely conclusive model and ruling out all others is probably impossible. However, a more thorough theoretical discussion is needed, considering other models and arriving at one that fits the data (even if it may not be the only one that fits the data).Finally, the reviewers make concrete suggestions to focus the manuscript on the most interesting aspects of this study worth developing further: a more detailed quantitative analysis of the CD4/CD8 and TCRα/TCRβ repertoires, the reported differences between TCRα and TCRβ, and the differing role of generation probability.

We carefully read your input and value your effort to make excellent suggestions for additional analyses and interpretation, and for providing a more complete context of previous studies.

We have carried out a major revision of the text, and have added a number of additional analyses to address the comments of the editors and reviewers. Our point-by-point response to the reviewers’ comments is detailed below. As you point out, while analysis of large clones has been performed previously, and various hypotheses for their origin have been advanced (we have added a much more complete review of this literature to the Introduction), careful comparison of different possible models has often been lacking, and analysis of the experimental data has sometimes been insufficiently rigorous in considering potential technical confounders.

We believe that our study has re-examined this question, and both by thorough analysis of new sets of experimental data and by comparison to mathematical models, has identified several different mechanisms which may give rise to observed abundant TCR chains in the naïve pool. Critically, we have been able to identify evidence for several different underlying mechanisms which may contribute to the observed data on TCR abundance, and distinguish carefully and quantitatively between experimental bias (e.g. FACS sorter contamination and repeated sampling of RNA from the same cell), and bona fide examples of multiple copies of the same TCR chain from different cells. We have also been able to examine the relative contribution of different generation probabilities, and large clones to the observation of abundant TCR sequences. We clearly show that the former play a major role for abundant α sequences, and the latter plays the major role for the smaller number of abundant β sequences. We show that our data are compatible with an underlying power law distribution of clone sizes, or a more general mixture model of rare and abundant clones. Finally, we provide evidence that abundant TCRs may arise both from early expansion of T cells during the initial formation of the repertoire (indicated by enrichment for TCR sequences without insertions), and by antigen experienced T cells with naïve phenotype (indicated by enrichment in TCR chains with known antigen-specific annotations).

Taken together, we think that this represents the most comprehensive analysis of observed abundant TCR chains in the naïve repertoire of humans, and as such will be of broad interest to many readers of this journal.

Reviewer #1:The manuscript by de Greef et al. presents a computational framework to infer the clone size distribution of naive T cells. The main conclusion from this work is that there are large clones in the naive repertoire, the emergence of which could not be explained by the neutral model.I think both methodology and data are interesting, but additional analysis is needed to address possible concerns outlined below.1) The first dataset authors use consists of TCRA and TCRB repertoires of FACS-sorted naive and memory cells. As the authors note, FACS is not precise, and a result, naive subpopulation could be contaminated with abundant memory clonotypes. It would be useful to quantify the extent of this contamination. One way to visualize this is to make scatterplots of concentrations of each clonotype (both overlapping and non-overlapping) in naive vs. memory repertoire. As a control, such plots could also be done for naive and memory subsets of different individuals: there should be little correlation between clone frequencies and small overlap, since there should be no contamination on FACS in this case.

We have plotted the count in memory versus naïve as the reviewer suggested. The CD4 and CD8 data show clear differences, which reflect the different TCR abundance distributions of the two populations. The CD4 memory contain very few TCR sequences present more than 1000 times and show little evidence of contamination. The CD8 memory contain quite a large number of TCR sequences present more than 1000 times; for these larger clones, there is clear evidence of contamination, as shown by a linear dependence between the count in naïve versus the count in memory. Based on this data, we estimate a rate of contamination of 0.1-1.5% from sorting for CD8 cells, and <0.05% for CD4 cells. In order to mitigate against a possible confounding due to this sorting, we have repeated all the analysis in the paper, using “cleaned” data in which all TCR sequences found in memory and naïve are removed. This dual analysis is now incorporated in several figures. The additional data processing step did not alter any of the major conclusions of the study. The plots discussed above, and a detailed consideration of potential sort contamination have been added to the first section of the Results (Figure 1A and B).

2) The end of subsection “Abundant TCR sequences are frequently shared between naive and memory populations, and are enriched for high VDJ recombination probabilities”:"if the overlap were the result of contamination only, the P(σ) of the [overlapping] sequences would be expected to reflect those of the memory subsets. Since the overlap is markedly enriched for high generation probabilities, most of it cannot be caused by contamination"However, green dashed line (which is P(σ) of overlapping sequences) on Figure 1A for β chains (especially for CD8 in volunteer 2) seems to be very close to red lines (probs for memory clonotypes), suggesting that most overlap observed in β repertoires is caused by the contamination between two cell samples on FACS. I suggest modifying the text to avoid contradiction. Also, I think it would be useful to add theoretical prediction for the distribution of generative probabilities of overlapping sequences between mem and naive. If both memory and naive seqs have the same distribution of probs P(σ), then for overlapping(=recombined twice) sequences it should be P(σ) squared, if there is no contamination on FACS. Another way to obtain a prediction for the probability of overlapping sequences in the absence of contamination during FACS is to plot P(σ) for naive sequences of volunteer 1 overlapping with memory of volunteer 2 (and vice versa).

We agree with the reviewer that the original text was confusing, and have rewritten it to clarify these points. The difference between α and β chains in this respect is one of the key messages of the paper, and needs to be crystal clear. We think the confusion has been caused by not clearly distinguishing between the generative probability of a TCR sequence, and the probability of observing a TCR sequence once, twice or more in a sample. As the reviewer correctly points out, the P(σ) distribution of the overlapping β chains is the same as that of the memory population. This does not, however imply they are derived from each other, but only that the overlapping β chains do not have unusually high P(σ), and thus this factor is unlikely to contribute to overlap. In the case of α chains, the overlapping sequences do have a larger P(σ), and so this factor needs to be considered in deciding on the mechanism of the overlap.

In terms of the theoretical distribution the reviewer discusses, the same distinction applies. The probability that a TCR sequence is actually found in two samples (i.e. overlap) depends on its frequency in the underlying population, which is dependent in turn on P(σ) among other factors. As an example, if a TCR sequence has a P(σ) of 10^-6 then there will be on average 1 such TCR sequence in every 10^6 cells (ignoring selection, etc.); the probability of seeing it twice in such a sample is given by the binomial probability, with a chance of success p = P(σ). As an illustration, the probability of seeing a TCR sequence with P(σ) 10^-6 in a sample of 10^6 cells at least twice is 0.08; the corresponding probability of seeing a TCR sequence with P(σ) 10^-10 is 10^-13 (essentially zero). Thus TCR α chains could occasionally theoretically turn up in memory and naïve repertoires independently, simply because of their high P(σ); this is much less likely for β chains.

We have substantially rewritten the manuscript to address this fundamental point, and have included the additional plot that the reviewer suggests of naïve from one individual overlapping with memory of another (Figure 1C).

3) In the Discussion section, authors suggest that abundant clones in naive compartment could correspond to naive-like antigen-experienced subpopulation. It might be interesting to look for TCR amino acid sequences of these clones in existing databases of antigen-specific TCRs, such as VDJdb and McPAS.

We thank the reviewer for this interesting suggestion, and matched the data from the subsampling experiment with the VDJdb database (see Figure 5D). Remarkably, and as implied by the reviewer, we do in fact find a considerable enrichment of annotated TCR sequences in the overlapping population. This is true both in the total repertoires, and in the repertoires “cleaned” of all sequences which are also found in memory. As such an enrichment was to be expected, this only provides some additional evidence that the abundant clones may be enriched for naïve-like antigen-experienced subpopulation, although we discuss alternative interpretations of this result. These results have been added to the final section of the Results section.

4) Authors made additional experiment with splitting naive cells into three parts before the mRNA extraction to avoid the potential noise introduced by variance in TCR expression by different cells. Is it possible to quantify, how much bias is removed by using this design? E.g., what happens if we computationally join these three independently sequenced replicates back together, and then randomly assign each of UMIs to one of 3 portions? Are there many clones with elevated TCR transcription levels and thus inflated counts in bulk repertoires? Is this variance in TCR expression different for α and β chains? I think answers for these questions would be interesting for many groups sequencing TCR repertoires with RNA-based technology.

This is an excellent suggestion, and we performed the requested permutation 10 times. We find that the true number of sequences found in multiple samples is much lower than after redistributing the samples (Figure 2—figure supplement 1). We quantify the difference in estimated number of abundant chains in Author response table 1. Interestingly, we observe that the differences occur for both TCRα and TCRβ data, but the effect is much larger for β chains. We estimate that over 75% of duplets within a sample are likely to be due to sampling multiple TCRβ RNA molecules from one cell, and about 25% for TCRα.

**Author response table 1. resptable1:** 

Fraction of TCR sequences with incidence >
	Data	Permutation	Fold difference	% due to RNA
CD4α	4.3%	5.5%	1.3x	21.6%
CD4β	0.8%	3.7%	4.6x	78.2%
CD8α	4.1%	6.3%	1.5x	34.0%
CD8β	0.6%	5.4%	8.5x	88.3%

This new finding is consistent with our previous publication where we document that T cells on average contain in the order of 300 β RNA molecules, and 100 α RNA molecules per cell (Oakes et al., 2017). Moreover, since the efficiency of our ligation protocol is in the order of 1-5%, repeat sampling of RNA from the same cell is quite likely. As the reviewer points out this highlights the importance of our additional step of taking a single blood sample, dividing it into three and then analyzing repertoire on all three subsamples separately. We thank the reviewer for this excellent suggestion and have added the permutation analysis results in the Materials and methods section (subsection “Subsampling to exclude inflated abundance through multiple RNA contributions by single cells” and Figure 2—figure supplement 1) and discussed the issue in detail in the main Results section.

5) Authors fit parameters for different clone size distributions using this additional experiment with splitting naive repertoire into three parts. It would be interesting to compare resulting best fit prediction for clone size distributions to clone size distributions observed in bulk naive repertories of two volunteers, which are analyzed in the first part of the paper (e.g. rank-frequency scatterplots for model vs. data).

We agree with the reviewer that this is an interesting analysis, and we have added it to the manuscript (Figure 4—figure supplement 1). As expected, the predictions under-estimate the number of doublets, especially from β chains, because of the impact of sampling multiple RNAs from the same cell as discussed in detail above (point 4). Apart from this, the model fits well to the “uncleaned” data in most cases, but the cleaned data under-represents abundant clones, suggesting that the clones shared between memory and naïve represent genuine abundant naïve clones. The only exception is for CD4 β, where we observe fewer large clones than the model predicts.

6) Another potential explanation for large naive clonotypes may be in the early repertoire development: it is known that TdT is not working, and thus first T-cells lack N-insertions. Previously our group have shown (see Figure 3 in Pogorelyy et al., 2017), that most abundant TCRβ from naive and cord blood (but not memory) repertoires are enriched with zero insertion clonotypes. It would be interesting to see, if abundant naive clonotypes observed in this study are also enriched with zero-insertion clonotypes (e.g. Figure 1B analog with some estimate of total number of N-insertions in each bin on y-axis).

We thank the reviewer for pointing out this important alternative explanation, performed the analysis, and added the results to the last part of the Results section. Interestingly, the more abundant TCRα sequences are indeed strongly enriched for zero insertion rearrangements, as the reviewer suggested. For TCRβ, this effect is smaller, but we still observe that abundant β chains (measured as incidence > 1) are enriched for zero insertions. So, these observations are consistent with a role for large clones that are created at or just before birth. This new and interesting observation is now included and discussed in the manuscript in Figure 5A, Figure 1—figure supplement 1 and Figure 5—figure supplement 5A.

Reviewer #3:This work investigates potential determinants of the distributions of naïve CD4 and CD8 T cell clone sizes using computational analysis and mathematical modelling of high-throughput TCRα and TCRβ sequence data.There are several major concerns about this manuscript:1) Although this work improves on previous studies by considering both TCRα and TCRβ sequences in naive CD4 and CD8 T cell populations and using potentially more accurate quantification of clone sizes (using UMIs), the main conclusion of the manuscript, that the naïve TCR repertoire has a broad distribution of clone sizes, is not substantially novel. Heterogeneous clone sizes in naïve T cell repertoires have been reported, and investigated, in many previous studies (e.g. Robins et al., 2009, Quigley et al., 2010, Venturi et al., 2011, Qi et al., 2014, Pogorelyy et al., 2017). Previous studies have also reported substantial overlap of TCR sequences between naïve and memory CD8^+^ T cell compartments that is enriched for high abundance TCR sequences that have limited numbers of n-additions (e.g. Robins et al., 2010, Venturi et al., 2011). Furthermore, many studies have previously investigated the associations between V(D)J recombination / TCR production probability, naïve TCR clone size and inter-individual sharing of TCRs (e.g. Robins et al., 2009, Robins et al., 2010, Quigley et al., 2010, Venturi et al., 2011, Li PNAS 2014, Pogorelyy et al., 2017). Importantly, this manuscript has not acknowledged this substantial body of previously published and highly relevant research.

We thank the reviewer for pointing out this body of literature and apologize for not referring to these publications. We now include a more thorough review of the previous literature in the Introduction.

2) While the various models of the naïve T cell clone sizes may be novel, they did not provide sufficient new insights into the primary mechanisms driving the naïve T cell distribution. This was largely due to the fact that no one model considered by the authors could fully explain the observed TCR clone size distributions. One possible reason for this is the growing evidence for developmental- and age-linked heterogeneity in naïve T cell populations (e.g. Hogan et al., 2015, Rane et al., 2018, Reynaldi et al., 2019). Although the authors show model results for a range of parameters, this analysis does not account for the potential impact on the adult T cell repertoire of changes in naïve T cell dynamics over the lifespan of an individual. For example, it has been recently suggested that high abundance zero n-addition TCRs in the adult naïve repertoire have survived since early development and their high abundance is due to different homeostatic pressures in the peripheral repertoire during early development (Pogorelyy et al., 2017). This previous relevant research has not been considered in this manuscript.

We completely agree with the reviewer (and with the very similar point made by reviewer 1). We have now added the analysis of zero n-additions as described above, and acknowledge the previous literature.

3) A potentially interesting conclusion in this study is the limited association between TCRβ abundance, TCRβ production probability, and TCRβ sharing. This result is not consistent with the findings of many previous studies focused on TCRβ repertoires (listed above for concern #1). However, this discrepancy with previous studies was not discussed in the manuscript. Moreover, the authors have not undertaken further investigation to determine the robustness of this result to various parameters/assumptions in the computational analysis or potential explanations for this discrepancy.

Previous studies showed that production probability of TCRα and TCRβ play a role in predicting abundance and sharing. Using IGoR and incidence rather than abundance, we confirm this for the TCRα sequences, and we suggest that this plays a much smaller role for TCRβ. The latter is the discrepancy that the reviewer is correctly pointing out.

We agree with previous results, as in our data, the TCRβ with incidence > 1 are also more public than TCRβ with incidence 1 (Figure 2—figure supplement 2) and the TCRβ with incidence 2 are enriched for high P(σ) (Figure 2B). At the same time, this confirmation shows that our results do not depend on our computational analysis. Moreover, the results are qualitatively similar when using two different pipelines (Decombinator and RTCR) and with and without cleaning of naïve sequences that overlap with memory samples. Our new finding is that in the naïve repertoire, there is a small population of very large clones (as measured with incidence 3), that are not explained by high generation probability. In line with this, these TCRβ sequences tend to be not public. Thus, there is no discrepancy, but only a new insight on very large naïve clones. We discuss this in detail in the manuscript.